# Bayesian parameter estimation for automatic annotation of gene functions using observational data and phylogenetic trees

**George G. Vega Yon**[1]*, **Duncan C. Thomas**[1], **John Morrison**[1], **Huaiyu Mi**[2], **Paul D. Thomas**[2], **Paul Marjoram**[1]

**1** Division of Biostatistics, Department of Preventive Medicine, University of Southern California, Los Angeles, California, United States of America, **2** Division of Bioinformatics, Department of Preventive Medicine, University of Southern California, Los Angeles, California, United States of America

* vegayon@usc.edu

## Abstract

Gene function annotation is important for a variety of downstream analyses of genetic data. But experimental characterization of function remains costly and slow, making computational prediction an important endeavor. Phylogenetic approaches to prediction have been developed, but implementation of a practical Bayesian framework for parameter estimation remains an outstanding challenge. We have developed a computationally efficient model of evolution of gene annotations using phylogenies based on a Bayesian framework using Markov Chain Monte Carlo for parameter estimation. Unlike previous approaches, our method is able to estimate parameters over many different phylogenetic trees and functions. The resulting parameters agree with biological intuition, such as the increased probability of function change following gene duplication. The method performs well on leave-one-out cross-validation, and we further validated some of the predictions in the experimental scientific literature.

## Author summary

Understanding the individual roles that genes play in life is a key issue in biomedical science. While information regarding gene functions is continuously growing, the number of genes with uncharacterized biological functions is still greater. Because of this, scientists have dedicated much of their time to build and design tools that automatically infer gene functions. One of the most promising approaches (sometimes called "phylogenomics") attempts to construct a model of inheritance and divergence of function along branches of the phylogenetic tree that relates different members of a gene family. If the functions of one or more of the family members has been characterized experimentally, the presence or absence of these functions for other family members can be predicted, in a probabilistic framework, based on the evolutionary relationships. Previously proposed Bayesian approaches to parameter estimation have proved to be computationally intractable, preventing development of such a probabilistic model. In this paper, we present a simple

**Data Availability Statement:** All the data can be downloaded from the pantherdb.org and geneontology.org websites. Furthermore,

processed data used is included in the paper's website https://github.com/USCbiostats/aphylo-simulations.

**Funding:** Research reported in this publication was fully funded by the Division of Cancer Epidemiology and Genetics of the National Institutes of Health under award number P01CA196596. The content is solely the responsibility of the authors and does not necessarily represent the official views of the National Institutes of Health. Computation for the work described in this paper was supported by the University of Southern California's Center for Advanced Research Computing (https://carc.usc.edu). The funders had no role in study design, data collection and analysis, decision to publish, or preparation of the manuscript.

**Competing interests:** The authors have declared that no competing interests exist.

model of gene-function evolution that is highly scalable, which means that it is possible to perform parameter estimation not only on one family, but simultaneously for hundreds of gene families, comprising thousands of genes. The parameter estimates we obtain coherently agree with what theory dictates regarding how gene-functions evolved. Finally, notwithstanding its simplicity, the model's prediction quality is comparable to other more complex alternatives. Although we believe further improvements can be made to our model, even this simple model makes verifiable predictions, and suggests areas in which existing annotations show inconsistencies that may indicate errors or controversies.

## 1 Introduction

The overwhelming majority of sequences in public databases remain experimentally uncharacterized, a trend that is increasing rapidly with the ease of modern sequencing technologies. To give a rough idea of the disparity between characterized and uncharacterized sequences, there are $\sim 15$ million protein sequences in the UniProt database that are candidates for annotation, while, only 81,000 (0.3%) have been annotated with a Gene Ontology (GO) term based on experimental evidence. It is therefore a high priority to develop powerful and reliable computational methods for inferring protein function. Many methods have been developed, and a growing number of these have been assessed in the two Critical Assessment of Function Prediction (CAFA) experiments held to date [1, 2].

In previous work, we developed a semi-automated method for inferring gene function based on creating an explicit model of function evolution through a gene tree [3]. This approach adopts the "phylogenetic" formulation of function prediction first proposed by Eisen [4], and the use of GO terms to describe function as implemented in the SIFTER software (Statistical Inference of Function Through Evolutionary Relationships) developed by Engelhardt et al. [5]. To date, our semi-automated method has been applied to over 5000 distinct gene families, resulting in millions of annotations for protein coding genes from 142 different fully sequenced genomes. However, this approach requires manual review of GO annotations, and manual construction of distinct models of gene function evolution for each of the 5000 families. Even using extensive curation and complex software, called PAINT, the semi-automated inference process has taken multiple person-years. Further, the semi-automated process cannot keep up with the revisions that are constantly necessary due to continued growth in experimentally supported GO annotations.

Here, we describe an attempt to develop a fully automated, probabilistic model of function evolution, that leverages the manually constructed evolutionary models from PAINT, for both training and assessing our new method. Related work has previously been undertaken in this area, and a probabilistic framework for function prediction has been implemented in SIFTER. However, this framework assumes a model of function evolution that limits its applicability in function prediction. First, it was developed specifically only to treat molecular function, and not cellular component and biological process terms that could, in principle, be predicted in a phylogenetic framework. Cellular component and biological process GO terms are the most commonly used in most applications of GO [6], so these are of particular practical importance. And second, while it has proven to provide good predictions (about 73% accuracy using the area under the curve statistic), it cannot be scaled and the model itself provides no theoretical insights whatsoever. In practice, perhaps the most serious problem for any inference method is the sparseness of experimental annotations compared to the size of the tree. As a result, standard parameter estimation techniques for the SIFTER framework are impossible, and

consequently for SIFTER2.0, the transition matrix parameters are fixed at somewhat arbitrary values [7].

In order to overcome these problems, we propose a much simpler evolutionary model, in which (like the semi-automated PAINT approach) each function is treated as an independent character that can take the value 1 (present) or 0 (absent) at any given node in the phylogenetic tree. Using information about experimental annotations available in the GO database, and phylogenetic trees from the PANTHER project [8], we show that we can build a parsimonious and highly scalable model of functional evolution that provides both intuitive insights on the evolution of functions and reasonably accurate predictions.

We emphasize that in this paper we focus on developing a model in which parameter estimation is feasible not only over a single gene family, but jointly over multiple gene families. Parameter estimation in SIFTER was attempted for only a single, specific gene family (AMP/ adenosine deaminase), and was otherwise infeasible due to multiple factors, including the use of a transition-based function evolution model that results in a combinatorially increasing number of parameters as the number of different functions increases. Further work on SIFTER [7] developed an approximation to the likelihood calculation, and did not attempt parameter estimation, instead relying on fixed parameters.

Our goal in this paper is to provide a method that retains computational tractability when estimating the evolutionary parameters, and does not therefore require the use of fixed parameter estimates. Indeed, by adopting a Bayesian perspective we allow for uncertainty in parameter estimates by integrating out over the posterior distribution for those parameter values. In order to do so, we simplify the evolutionary model to treat gain and loss of each function independently, minimizing the model parameters. We consider only two properties of each gene family tree: topology and branch type, distinguishing only between branches following speciation versus gene duplication events. Thus, by simplifying the model we regain computational tractability on large trees without paying any great price in terms of prediction error (see Results). This paves the way for future more refined versions of our model that add more realism to the evolutionary model.

The content of the paper is as follows: Section 2 presents mathematical notation and formally introduces the model, including parameter estimation and calculation of posterior probabilities. Section 3 presents a large-scale application in which we take a sample of annotations from the GO database along with their corresponding phylogenies, fit our model, and analyze the results. Finally, section 4 discusses method limitations and future research directions for this project.

## 2 Methods

In general terms, we propose a probabilistic model that reflects the process by which gene functions are propagated through an evolutionary tree. The fundamental idea is that for any given node in the tree, we can write down the probability of observing a function to be present for the gene as a function of model parameters and the functional state of its parent node, essentially modeling probability of gaining and losing function.

The baseline model has 5 parameters: the probability that the most recent common ancestor of the entire tree, i.e. the root node, had the function of interest, the probability of functional gain (the offspring node gains the function when the parent node did not have it), the probability of functional loss (the offspring node loses the function when the parent node had it), and two additional parameters capturing the probability that the gene was incorrectly labeled, i.e. the probability of classifying an absent function as present and vice-versa. We also

consider two simple extensions: specifying the functional gain and loss by type of evolutionary event (speciation or duplication), and pooling information across trees.

As explained later, in this version of our model, if there are multiple gene functions of interest, we analyze one function at a time (i.e., we treat those functions as independent). But later we also show results for a joint analysis of multiple functions, assuming they share parameter values. We then discuss further extensions to our model in the Discussion section.

We assume that our starting data consists of a set of gene annotations for a given tree. We further assume that those annotations occur only at the leaves of the tree (as is typical) and that those leaves were annotated via manual curation derived from experimental data (and are therefore subject to the misclassification probabilities outlined above). Our goal is then to predict the functional state of un-annotated leaves (and, conceivably, to assess the likely quality of the existing annotations). Our perspective is Bayesian. Therefore, we proceed by estimating the posterior distributions of the parameters of the evolutionary model and then, conditional on those estimated distributions, computing the posterior probability of function for each node on the tree.

We now give full details, beginning with some notation.

## 2.1 Definitions—Annotated Phylogenetic Tree

A phylogenetic tree $\Lambda \equiv (\mathcal{N}, \mathcal{E})$ is a tuple of nodes $\mathcal{N}$, and edges $\mathcal{E} \equiv \{(n, m) \in \mathcal{N} \times \mathcal{N} : n \mapsto m\}$ defined by the binary operator $\mapsto$ *parent of*. We define $\mathbf{Off}(n) \equiv \{m \in \mathcal{N} : (n, m) \in \mathcal{E}, n \in \mathcal{N}\}$ as the set of offspring of node $n$, and $\mathbf{p}(m) \equiv \{n \in \mathcal{N} : (n, m) \in \mathcal{E}, m \in \mathcal{N}\}$ as the parent node of node $m$. Given a tree $\Lambda$, the set of leaf nodes is defined as $L(\Lambda) \equiv \{m \in N: \mathbf{Off}(m) = \{\emptyset\}\}$.

Let $\mathbf{X} \equiv \{x_n\}_{n \in \mathcal{N}}$ be a vector of annotations in which the element $x_n$ denotes the state of the function at node $n$, taking the value 1 if such function is present, and 0 otherwise. We define an Annotated Phylogenetic Tree as the tuple $D \equiv (\Lambda, \mathbf{X})$.

Our goal is to infer the true state of $\mathbf{X}$, while only observing an imperfect approximation of it, $\mathbf{Z} = \{z_l\}_{l \in \mathcal{N}}$, derived from experimentally supported GO annotations [6]. Typically only a small subset of the leaf nodes will have been annotated. We refer to the tuple $\tilde{D} \equiv (\Lambda, \mathbf{Z})$ as an Experimentally Annotated Phylogenetic Tree. Finally, let $\tilde{D}_n \subset \tilde{D}$ be the induced experimentally annotated subtree that includes all information—i.e., tree structure and observed annotations – regarding the descendants of node $n$ (including node $n$ itself). This object constitutes a key element of the likelihood calculation. Table 1 summarizes the mathematical notation.

**Table 1. Mathematical notation.**

| Symbol | Description |
|---|---|
| $\Lambda \equiv (\mathcal{N}, \mathcal{E})$ | Phylogenetic Tree. |
| $\mathbf{p}(n)$ | Parent of node $n$. |
| $\mathbf{Off}(n)$ | Offspring of node $n$. |
| $\mathbf{X} \equiv \{x_n\}_{n \in \mathcal{N}}$ | True annotations. |
| $\mathbf{Z} \equiv \{z_n\}_{n \in \mathcal{N}}$ | Experimental annotations. |
| $D \equiv (\Lambda, \mathbf{X})$ | Annotated Phylogenetic Tree. |
| $\tilde{D} \equiv (\Lambda, \mathbf{Z})$ | Experimentally Annotated Phylogenetic Tree. |
| $\tilde{D}_n$ | Induced Experimentally Annotated Subtree of node $n$. |
| $\tilde{D}_n^c$ | Complement of $\tilde{D}_n$. |

## 2.2 Likelihood of an Annotated Phylogenetic Tree

**2.2.1 Baseline model.** Our evolutionary model for gene function is characterized by the phylogeny and five model parameters: the probability that the root node has the function, $\pi$, the probability of an offspring node either gaining or losing a function, $\mu_{01}$ and $\mu_{10}$, and the probability that either an absent function is misclassified as present or that a present function is misclassified as absent, $\psi_{01}$ and $\psi_{10}$ respectively, in the input data.

To simplify notation, we will write $\psi = (\psi_{01}, \psi_{10})$ and $\mu = (\mu_{01}, \mu_{10})$ when referring to those pairs. Table 2 summarizes the model parameters.

Following [9], we use a pruning algorithm to compute the likelihood of an annotated phylogenetic tree. In doing so, we visit each node in the tree following a post-order traversal search, i.e., we start at the leaf nodes and follow their ancestors up the tree until we reach the root node.

Given the true state of leaf node $l$, the mislabeling probabilities are defined as follows:

$$\mathbb{P}(z_l = 1 | x_l = 0) = \psi_{01}, \quad \mathbb{P}(z_l = 0 | x_l = 1) = \psi_{10}$$

We can now calculate the probability of leaf $l$ having state $z_l$, given that its true state is $x_l$ as:

$$\mathbb{P}(z_l | x_l, \psi) = \begin{cases} \psi_{01} z_l + (1 - \psi_{01})(1 - z_l), & \text{if } x_l = 0 \\ \psi_{10}(1 - z_l) + (1 - \psi_{10}) z_l, & \text{if } x_l = 1 \end{cases} \tag{1}$$

Similarly, the functional gain and functional loss probabilities are defined as follows:

$$\mathbb{P}(x_m = 1 | x_{\mathbf{p}(m)} = 0) = \mu_{01}, \quad \mathbb{P}(x_m = 0 | x_{\mathbf{p}(m)} = 1) = \mu_{10}$$

Note that in this version of our model we assume that these probabilities are independent of the time that has passed along the branch connecting the two nodes. We return to this point in the Discussion. Now, for any internal node $n \in \mathcal{N}$, we can calculate the probability of it having state $x_n$, given the state of its parent $\mathbf{p}(n)$ and the vector of parameters $\mu$, as:

$$\mathbb{P}(x_m | x_{\mathbf{p}(m)}, \mu) = \begin{cases} \mu_{01} x_m + (1 - \mu_{01})(1 - x_m) & \text{if } x_{\mathbf{p}(m)} = 0 \\ \mu_{10}(1 - x_m) + (1 - \mu_{10}) x_m & \text{if } x_{\mathbf{p}(m)} = 1 \end{cases} \tag{2}$$

Together with (1), and following [9], this allows us to calculate the probability of the interior node $n$ having state $x_n$ conditional on $\tilde{D}_n$, its induced subtree, as the product of the conditional probabilities of the induced subtrees of its offspring:

$$\mathbb{P}(\tilde{D}_n | x_n, \psi, \mu) = \prod_{m \in \mathbf{Off}(n)} \mathbb{P}(\tilde{D}_m | x_n), \tag{3}$$

**Table 2. Model parameters.**

| Parameter | Description |
|---|---|
| $\pi$ | Probability that the root node has the function |
| $\mu_{01}$ | Probability of gaining a function |
| $\mu_{10}$ | Probability of losing a function |
| $\psi_{01}$ | Probability of experimental mislabeling of a 0 |
| $\psi_{10}$ | Probability of experimental mislabeling of a 1 |

where

$$\mathbb{P}(\tilde{D}_m | x_n) = \begin{cases} \sum_{x_m \in \{0,1\}} \mathbb{P}(\tilde{D}_m | x_m, \psi, \mu) \mathbb{P}(x_m | x_n, \mu) & \text{if } m \text{ is interior}, \\ \sum_{x_m \in \{0,1\}} \mathbb{P}(x_m | z_m, \psi) \mathbb{P}(x_m | x_n, \mu) & \text{if } m \text{ is leaf}. \end{cases}$$

This is a recursive function that depends upon knowing the offspring state probabilities for internal nodes, which, since we are using a pruning algorithm, will already have been calculated as part of the process. Finally, the probability of the experimentally annotated phylogenetic tree can be computed using the root node conditional state probabilities:

$$\mathbb{P}(\tilde{D} | \psi, \mu, \pi) = \sum_{x_0 \in \{0,1\}} \mathbb{P}(x_0 | \pi) \mathbb{P}(\tilde{D} | x_0, \psi, \mu) \tag{4}$$

where $\mathbb{P}(x_0 | \pi) = \pi x_0 + (1 - \pi)(1 - x_0)$.

In the next section we discuss the computational complexity of this method, and introduce additional refinements that allow us take into account prior biological knowledge that might constrain the parameter space and alleviate the typical sparseness of the curated data—we return to these issues in the Discussion.

**2.2.2 Computational complexity.**   As the core of the probability function is an application of Felsenstein's pruning algorithm and, under our model, functions evolve independently, the complexity of this algorithm is linear in the number of leaves and functions, $O(\# \text{ of leaves} \times \#$ of functions). Furthermore, given that un-annotated leaves do not contribute to the overall probability of annotations, we go a step further and calculate this probability using a reduced pruning sequence, namely, that of the induced subtree in which only nodes that can be traced to a descendant with an annotation are kept. Considering the fact that experimental annotations are typically very sparse, using the reduced pruning sequence greatly accelerates the statistical inference.

From the practitioner's point of view, users can perform large scale analyses, including both parameter estimation and subsequent function prediction, using a single computational thread and less than one gigabyte of RAM, *i.e.*, no more than a "regular" computer. As we will illustrate in the Results section, we were able to perform parameter estimation using Markov Chain Monte Carlo on a dataset including $\sim$1,300 annotated proteins within 5 minutes, and the subsequent calculation of posterior probabilities for all leaves, annotated or not, about 83,000 of them in total, in roughly one second, using only one computational thread.

**2.2.3 Pooled data models.**   As mentioned earlier, the sparsity of experimental annotations typically observed in this context makes inference challenging. However, in order to improve inference we might attempt to "borrow strength" across a set of functions, by combining them in a single analysis. As a "proof of principle" of this idea, we will also show results in which we assume that sets of annotated phylogenetic trees share population parameters. This way, we can estimate a joint model by pooling those annotated phylogenetic trees, providing stronger inference about the evolutionary parameters and therefore, we hope, more accurate inference of gene annotations. We note that we have strived to make sure our software implementation extremely computationally efficient, allowing us to estimate a joint model with hundreds of annotated trees in reasonable time on a single machine.

Our results show that using a pooled-data model for parameter estimation greatly increases the model's predictive power. In the Discussion we will outline possible future, less simplified, approaches to this problem.

**2.2.4 Type of evolutionary event.**   The non-leaf nodes in the phylogenetic trees that we are considering here come in two types, reflecting duplication and speciation events. It has

been widely-observed that change of gene function most commonly occurs after duplication event (broadly speaking, the extra copy of a gene is free to change function because the original copy is still present.) Speciation events, on the other hand, are mostly driven by external causes that do not necessarily relate to functional changes in one specific gene, meaning that, while we may observe some differences between siblings, their function generally remains the same.

Our model reflects this biological insight by allowing the functional gain and loss probabilities to differ by type of event. In particular, instead of having a single parameter pair $(\mu_{01}, \mu_{10})$, we now have two pairs of functional gain and loss parameters, $(\mu_{01d}, \mu_{10d})$ and $(\mu_{01s}, \mu_{10s})$, the gain and loss probabilities for duplication and speciation events, respectively.

## 2.3 Estimation and prediction

We adopt a Bayesian framework, introducing priors for the model parameters. In particular, as described in section 3, we use Beta priors for all model parameters. Estimation is then performed using Markov chain Monte Carlo [MCMC] using an Adaptive Metropolis transition kernel [10, 11] with reflecting boundaries at [0, 1] (see, for example, [12]) as implemented in the *fmcmc* R package [13]. We do this using the package *aphylo*, using the R programming language [14], which we have developed to implement this model (https://github.com/USCbiostats/aphylo). The R package, which fully integrates with the ape R package [15], also allows estimating model parameters using Maximum Likelihood Estimation [MLE] and Maximum A Posteriori estimates [MAP].

Regarding model predictions, once we fit the model parameters we use the calculated probabilities during the computation of the likelihood function (pruning process) and feed them into the posterior probability prediction algorithm, which is exhaustively described in Section 5.1 (see also [16]). It is important to notice that, for model assessment, predictions are made using a leave-one-out approach, meaning that to compute the probability that a given gene has a given function, we remove the annotation for that gene from the data before calculating the overall likelihood of the tree. Otherwise we would be including that gene's own observed annotation when predicting itself. The latter point is relevant for evaluating prediction error. Prediction of unannotated genes, on the other hand, is performed using all the available information.

In general, whenever using MCMC to estimate the model parameters, we used 4 parallel chains and verified convergence using the Gelman-Rubin [11] statistic. In all cases, we sampled every tenth iteration after applying a burn-in of half of the sample. As a rule of thumb, the processes were considered to have reached a stationary state once the potential scale reduction factor statistic was below 1.1. When we report point estimates for parameters we use the mean value across sampled iterations for all chains after burn-in.

To measure prediction quality, we use the Mean Absolute Error [MAE], which we calculate as follows:

$$\text{MAE} = |\mathbf{Z}|^{-1} \sum_n |z_n - \hat{x}_n| \tag{5}$$

Where $|Z|$ is the number of observed annotations, $z_n$ is the observed annotation (0/1), and $\hat{x}_n$ is the predicted probability that the $n$-th observation has the function of interest. Since $z_n$ is binary, perfect predictions result in a score of 0, completely wrong predictions result in a score of 1, and a random coin toss results in a score of 0.5. While other statistics, such as accuracy and Area Under the Receiver Operating Characteristic Curve [AUC] (see [17] for a general overview), are most commonly used for binary prediction models, we prefer to use MAE since it is: (a) robust to unbalanced data, (b) probabilistic and thus does not depend on choice of

threshold for prediction, and (c) robust to probabilistic noise, see [18, 19] for a more general discussion.

We now turn our attention to an application using experimental annotations from the Gene Ontology Project.

# 3 Results

To evaluate performance of our model, we used data from the Gene Ontology project [6], release 2016-11-01 together with PANTHER version 15.0 [20], which includes about 15,000 reconciled trees reconstructed with the GIGA algorithm [21], modified to include horizontal transfer inference as described in [22]. In particular, in order to assess the potential utility of combining information across similar genes, we used our model in two different ways: a pooled-data model, in which we treated all trees as if they had the same evolutionary parameter values, and a one-tree-at-a-time model, in which we estimated parameters, and then imputed functional status, for each tree separately.

Since in our model functions are assumed to evolve independently, trees having multiple GO terms were treated as multiple trees having a single function each. To validate our predictions, we opted for analyzing the data using leave-one-out cross-validation. As a result, we only kept those trees in which there were at least two genes with a positive annotation, "present", and two others with a negative annotation, "absent." This yielded a total of 1,491 experimental annotations on 1,325 genes, grouped in 114 PANTHER trees; which once they were expanded to have one function per tree, resulted in 138 trees.

A key feature of the resulting dataset is that most leaves still have no annotation, as experimental evidence is still very sparse. Furthermore, in order to test the robustness of our results, we compared 4 analyses: MCMC with a uniform prior for parameter values, MCMC with a Beta prior (for both of which we report the mean,) Maximum Likelihood Estimation [MLE], and Maximum A Posteriori estimates [MAP]. The shape parameters used for MCMC and MAP with Beta priors, which are detailed in Table 3, were chosen to reflect the biological intuition that change of state probabilities should be higher at duplication events.

Finally, prediction quality was measured using leave-one-out cross-validated Mean Absolute Error [MAE] and Area Under the Curve Receiver Operating Characteristic [AUC].

## 3.1 Pooled-data model

First we show results for the pooled-data model, in which we combine the analysis of multiple trees, assuming parameter values do not vary between trees. The resulting parameter estimates are presented in Table 4, and are seen to reflect biological intuition: we see high probabilities of change of function at duplication events, which occurs regardless of whether we use an informative prior or not. While both probabilities are high, gain of function is roughly twice as likely as loss of function at such nodes. We note that we estimate relatively high values of $\psi_{01}$ and low values for $\psi_{10}$. We believe this is largely due to the sparsity of negative annotations, i.e.

**Table 3. Parameters for the beta priors used in MCMC and MAP estimation.** Shape parameters $(\alpha, \beta)$ were set with the prior that functional gain and loss, $(\mu_{01d}, \mu_{10d})$, are more likely at duplication events.

| Parameter | $\alpha$ | $\beta$ | Mean $\alpha/(\alpha + \beta)$ |
|---|---|---|---|
| $\psi_{01}, \psi_{10}$ | 2 | 9 | 0.18 |
| $\mu_{01d}, \mu_{10d}$ | 9 | 2 | 0.81 |
| $\mu_{01s}, \mu_{10s}$ | 2 | 9 | 0.18 |
| $\pi$ | 2 | 9 | 0.18 |

**Table 4. Parameter estimates for model with experimentally annotated trees.** Overall, the parameter estimates show a high level of correlation across methods. Regardless of the method, with the exception of the root node probabilities, most parameter estimates remain the same. As expected, mutation rates at duplication nodes, indexed with a *d*, are higher than those observed in speciation nodes, indexed with an *s*.

| | (1) | (2) | (3) | (4) |
|---|---|---|---|---|
| Mislabeling | | | | |
| $\psi_{01}$ | 0.23 | 0.30 | 0.22 | 0.29 |
| $\psi_{10}$ | 0.01 | 0.01 | 0.00 | 0.01 |
| Duplication events | | | | |
| $\mu_{d01}$ | 0.97 | 0.95 | 1.00 | 0.97 |
| $\mu_{d10}$ | 0.53 | 0.63 | 0.52 | 0.61 |
| Speciation events | | | | |
| $\mu_{s01}$ | 0.05 | 0.06 | 0.05 | 0.06 |
| $\mu_{s10}$ | 0.01 | 0.02 | 0.01 | 0.02 |
| Root node | | | | |
| $\pi$ | 0.78 | 0.43 | 0.81 | 0.47 |
| Method | MCMC | MCMC | MLE | MAP |
| Prior | Uniform | Beta | - | Beta |
| AUC | 0.76 | 0.74 | 0.76 | 0.75 |
| MAE | 0.35 | 0.37 | 0.34 | 0.36 |

"*absent*", experimental annotations, which means that high values for $\psi_{10}$ are implausible and implies that any mislabelings must perforce be that of assigning function falsely. Thus we think the relatively high estimates of $\psi_{01}$ are an artefact that will likely disappear in the presence of less sparse annotation data. Along with the consistency of the parameter estimates, MAEs and ACUs are shown to be very similar as well. Fig 1 shows the corresponding Receiver Operating Characteristic plot for all four methods.

We note that parameter estimates are consistent across estimation methods. Here we used both non-informative (uniform) and relatively informative priors. We see that choice of prior had no significant effect on the final set of estimates, meaning that these are driven by data and not by the specified priors. The only exception is for the root node, $\pi$, which is hard to estimate since the root node exists far back in evolutionary time [23]. We also conducted a simulation study across a broader range of parameter values, using simulated datasets, in which we saw similar behavior. These results are shown in Section 5.2. However, we note that when analyzing one tree at a time, this robustness is lost [results not shown] because the sparsity of data typical in any single given tree does not permit strong inference regarding evolutionary parameters. Therefore, the prior becomes more influential.

We note that each instance of the estimation process (reaching stationary distribution in MCMC or finding the maxima in MLE and MAP) took between 1 to 3 minutes using a single thread. Furthermore, calculating the posterior probabilities for all leaves involved, which added up to about 83,000, took roughly one second. To put this number into context, we use the time calculator provided by SIFTER [5, 7], which is available at http://sifter.berkeley.edu/complexity/, as a benchmark. Since, as of today, SIFTER is designed to be used with one tree at a time, we compared our run time with the estimated run time for SIFTER for a tree with 590 leaves (the average size in our data). In such case, they estimate their algorithm would take about 1 minute to complete, with an upper bound of ~9 minutes, which translates into about 139 minutes to run on our entire dataset (upper bound of ~21 hours.) However, in SIFTER's defense, we note that their model allows for a greater level of generality than does our current

## Receiver Operating Characteristic

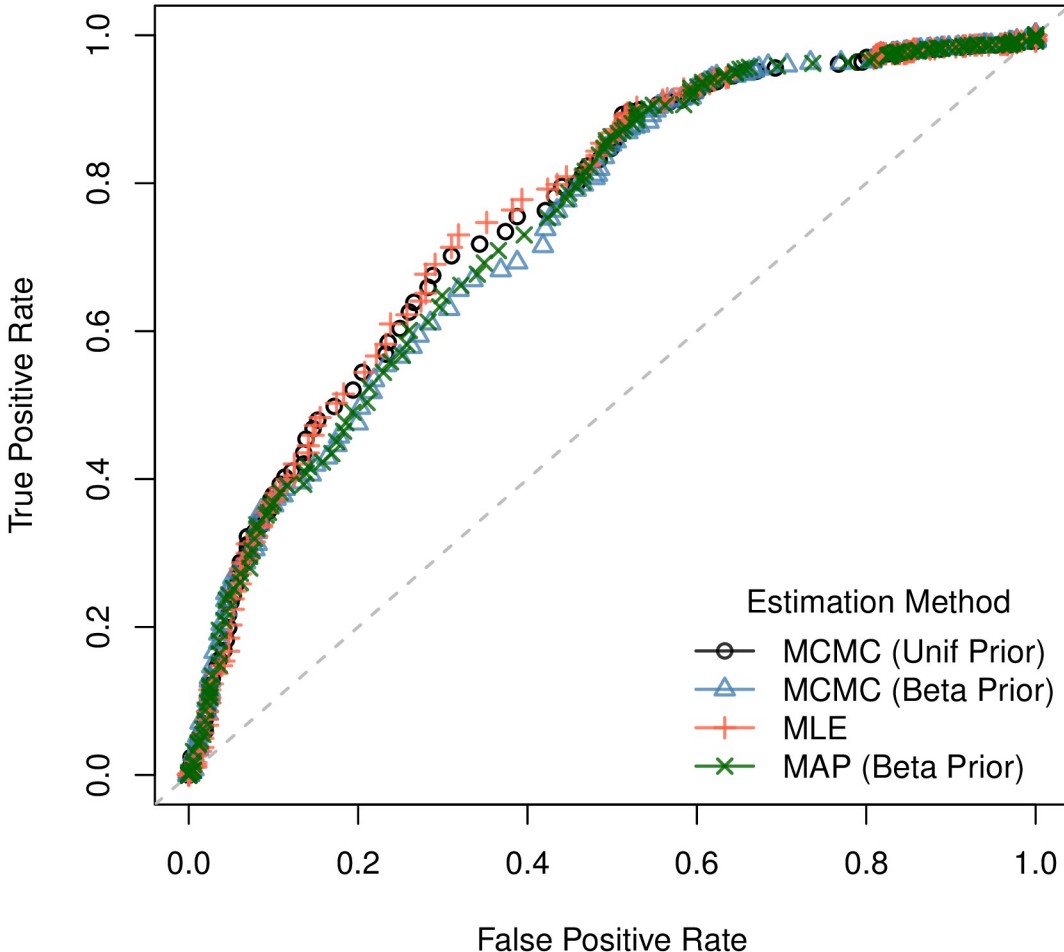

**Fig 1. ROC curve for each estimation method.** As reflected in the parameter estimates, ROC curves are very similar across all four methods.

implementation of our own method. Specifically, they allow for correlated changes across multiple gene functions. In Section 3.5 we provide a detailed analysis comparing our predictions with those of SIFTER.

### 3.2 One-tree-at-a-time

The joint analysis of trees we describe in the previous section makes the enormous assumption that parameter values do not vary across trees. This assumption is made for pragmatic reasons: while it is not likely to be correct (and more general approaches will be described in the Discussion), it is a simple analysis to implement and it allows us to quickly assess a baseline for the improvement in overall prediction quality that might result from performing inference jointly across trees. So, in this section we report results of a one-at-a-time analysis of gene function and assess whether the pooled model outperforms a one-at-a-time approach with regards to prediction quality measured as MAE, even using our extremely restrictive assumptions.

**Table 5. Differences in Mean Absolute Error [MAE].** Each cell shows the 95% confidence interval for the difference in MAE resulting from MCMC estimates using different priors (row method minus column method). Cells are color coded blue when the method on that row has a significantly smaller MAE than the method on that column; Conversely, cells are colored red when the method in that column outperforms the method in that row. Overall, predictions calculated using the parameter estimates from *pooled-data* predictions outperform *one-at-a-time*.

| | *Pooled-data* | One-at-a-time | |
| --- | --- | --- | --- |
| | Beta prior | Unif. prior | Beta Prior |
| *Pooled-data* | | | |
| Unif. prior | [-0.02,-0.01] | [-0.13,-0.10] | [-0.05,-0.02] |
| Beta prior | - | [-0.11,-0.08] | [-0.03,-0.01] |
| *One-at-a-time* | | | |
| Unif. prior | - | - | [0.06, 0.09] |

For each phylogeny, we fitted two different models using MCMC, one with an uninformative uniform prior and another with the same beta priors used for the pooled-data models, this is, beta priors with expected values close to zero or one, depending on the parameter. Again, as before, model predictions were undertaken in a leave-one-out approach and corresponding MAEs were calculated for each set of predictions. Table 5 shows the differences in Mean Absolute Error [MAE] between the various approaches.

From Table 5 we can see three main points: First, in the case of the one-at-a-time predictions, informative priors have a significantly positive impact on prediction error. Compared to the predictions using beta priors, predictions based on uniform priors have a significantly higher MAE with a 95% confidence interval [c.i.] ranging [0.06, 0.09]. Second, predictions based on the pooled-data estimates outperform predictions based on one-at-a-time parameter estimates, regardless of the prior used. And third, in the case of predictions based on pooled-data estimates, model predictions based on the parameter estimates using a uniform prior have significantly smaller MAE than predictions based on parameter estimates using a beta prior with a 95% c.i. equal to [-0.02, -0.01].

Given the sparsity of experimental annotation on most trees, a natural question to ask is: How important is it to have good quality parameter estimates in order to achieve good-quality predictions of gene function? In the next section we perform a sensitivity analysis to address this question.

## 3.3 Sensitivity analysis

Taking advantage of the computational efficiency that our method provides, we now present an exercise to illustrate a possible application of our method that may not be feasible to conduct for other models: a sensitivity analysis.

We analyze how the distribution of the Mean Absolute Error [MAE] corresponding to the 138 trees+functions just analyzed, which involved about 1,300 proteins, changes when posterior probabilities are calculated using a "wrong" evolutionary parameter values. In particular, each block of boxplots in Fig 2 shows a *ceteris paribus* type of analysis, this is, MAE as a function of changing one parameter at a time while fixing the others.

In the case of the mislabeling probabilities, sub-figures A1 and A2, it is not until they reach values close to 1 that the prediction error significantly increases. While higher values of $\psi_{10}$ have a consistent negative impact on MAE, the same is not evident for $\psi_{01}$. The latter could be a consequence of the low prevalence of 0 ("absence of function") annotations in the data, as illustrated in Fig 3.

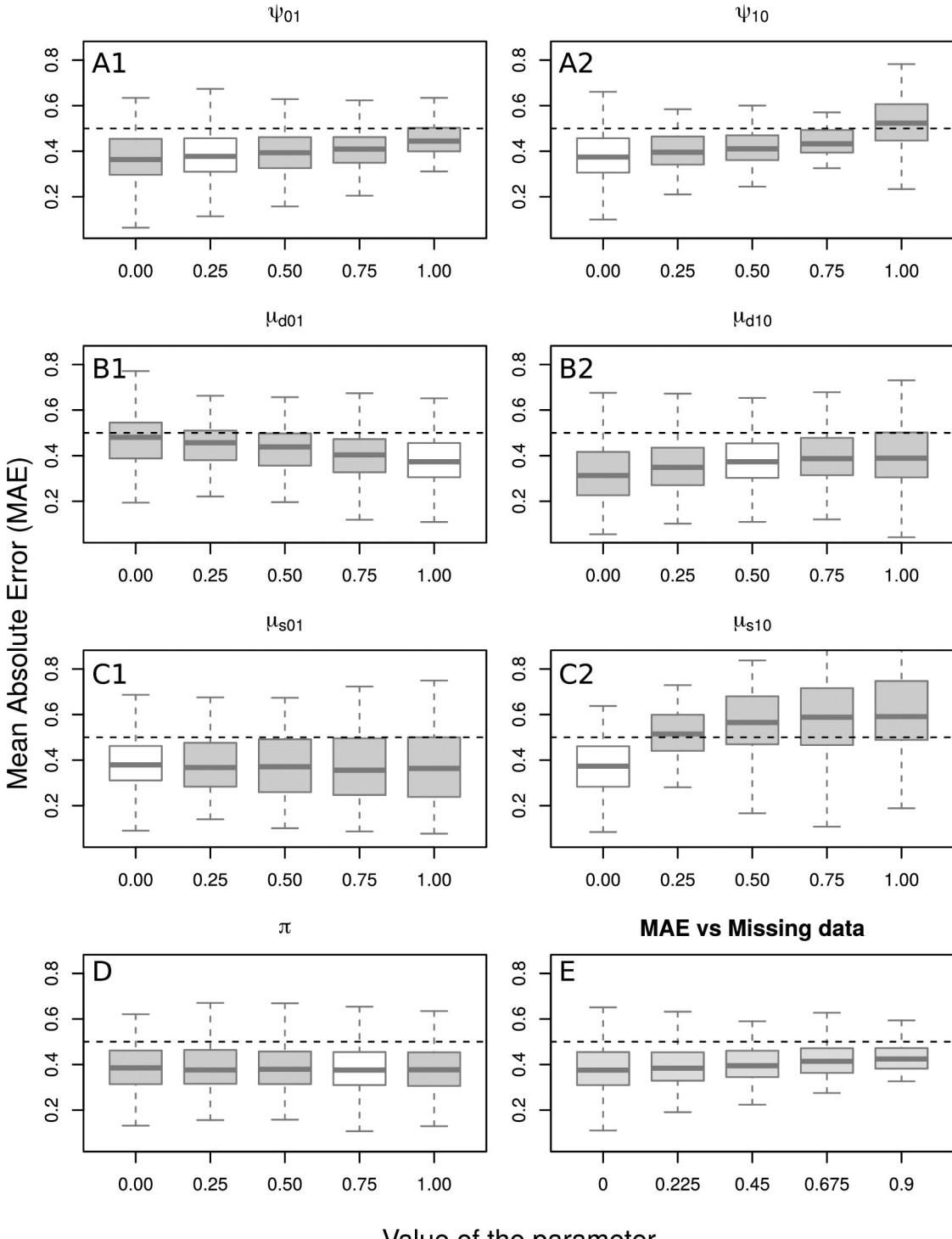

**Fig 2. Sensitivity analysis.** Boxplots of MAEs as a function of single parameter updates. Sub-figures A1 through D show how the MAEs change as a function of fixing the given parameter value ranging from 0 to 1. In each of these plots, the white box indicates the parameter value used to generate the data (i.e., the "correct" parameter value). The last boxplot, sub-figure E, shows the distribution of the MAEs as a function of the amount of available annotation data, that is, how prediction error changes as we randomly remove annotations from the available data.

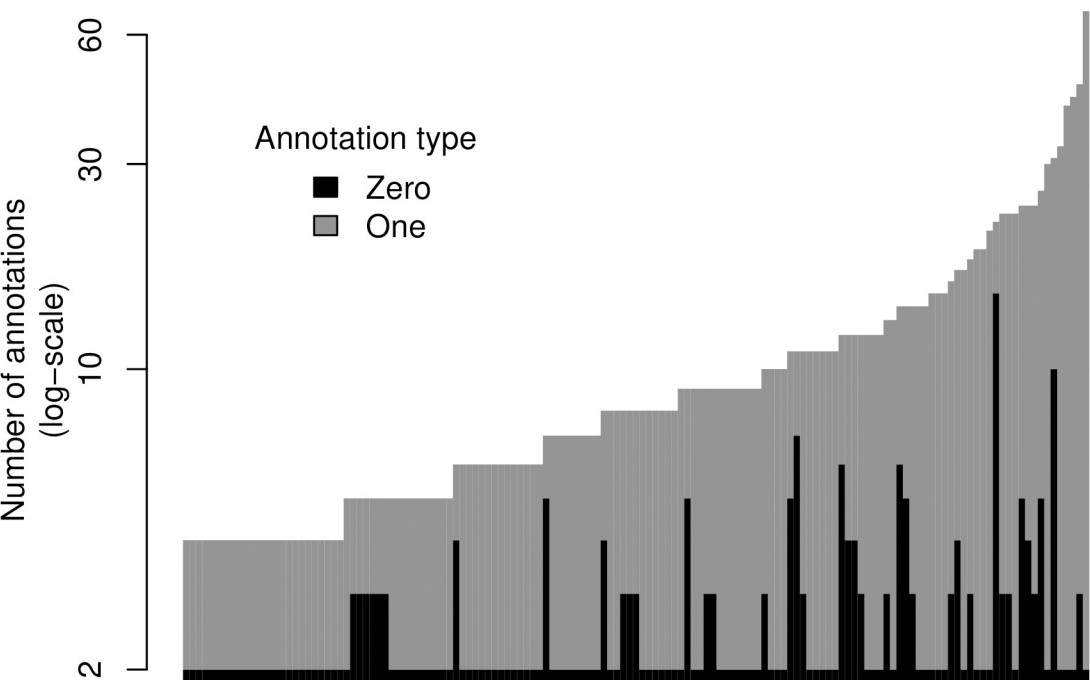

**Fig 3. Number of annotated leaves for all 138 trees by type of annotation.** Most of the annotations available in GO are positive assertions of function, that is, 1s. Furthermore, as observed in the figure, all of the trees used in this study have two "Not" (i.e., 0) annotations, which is a direct consequence of the selection criteria used, as we explicitly set a minimum of two annotations of each type per tree+function.

Functional gain and loss probabilities, sub-figures B1-B2 and C1-C2, respectively, have a larger effect on MAEs when misspecified, especially at speciation events, ($\mu_{s01}$, $\mu_{s10}$). This is largely because speciation events are much more common than duplication events across trees. Fig 4 shows a detailed picture of the number of internal nodes by type of event. Of the seven parameters in the model, the root node probability $\pi$ has a negligible impact on MAE.

We also show the impact of removing annotations (from trees already sparsely annotated) on MAEs in sub-figure E. While a significant drop may not be obvious from the last box in Fig 2, a paired t-test, Table 6, found a significant increase in MAE as more data was removed from the model. We note that we assume missingness "at random" in this analysis, rather than attempting to reflect some sort of non-random missingness that may sometimes be present in some data. Our reason for this is that, currently, annotation data is typically very sparse. This means that even if we tried to remove annotations in some clustered way, we will very rarely have two annotated leaves sufficiently close together to be affected by this choice. Thus, we expect results would be essentially identical in the vast majority of cases.

Extending this analysis, Fig 5 shows MAE as a function of number of annotations per tree. As expected, prediction is more accurate in trees with more annotations. It also appears that predictions on trees with proportionally fewer negative annotations out-performed those in trees with proportionally more negative annotations. A possible explanation for this is that the parameters used to compute the posterior probabilities were inferred using a dataset where the median tree had only two negative annotations.

Although useful as an example of the flexibility of our model, we note that the degree to which these conclusion generalize across other data and contexts remains to be seen. Our intent here is simply to showcase the flexibility of our model.

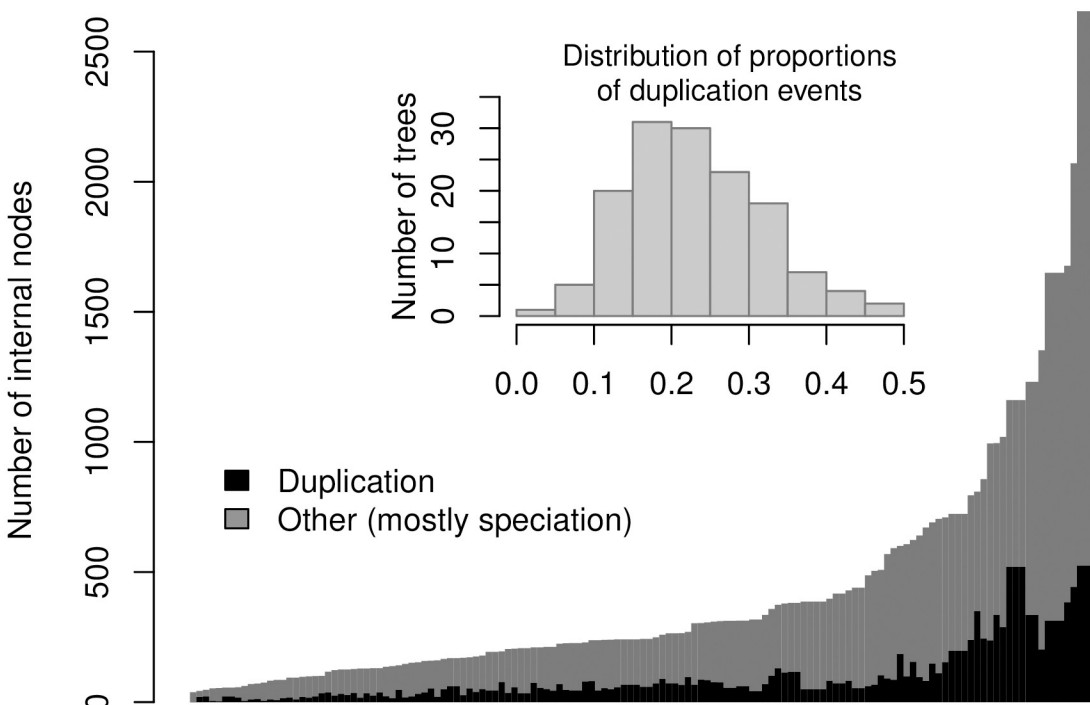

**Fig 4. Number of internal nodes by type of event.** Each bar represents one of the 138 trees used in the paper by type of event (mostly speciation). The embedded histogram shows the distribution of the prevalance of duplication events per tree. The minority of the events, about 20%, are duplications.

## 3.4 Featured examples

An important determinant of the quality of the predictions is how the annotations are co-located. For the model to be able to accurately predict a 1, it is necessary either that it follows immediately after a duplication event in a neighborhood of zeros (so that function can gained with reasonable probability), or that a group of relatives within a clade have the function (i.e., its siblings have the function). Inspired by a similar analysis of Revell [24], in which graphical methods are employed to analyze evolutionary traits, Figs 6 and 7 show examples of low and high MAE predictions respectively, illustrating how co-location impacts prediction error.

In the previous analyses, we reported predictions based on point estimates. It is more informative to leverage the fact that we have posterior distributions for parameter values. Hence, in this section, in which we are looking in detail at specific trees, we exploit the full posterior distribution, and generate credible intervals by repeatedly sampling parameter values from their

**Table 6. Effects of missing data on MAE on 138 trees.** The null hypothesis for the paired t-test is MAE(with X% annotations randomly dropped)—MAE(baseline) = 0. As more annotations are randomly removed from the tree, mean absolute error increases significantly.

| | Paired t-test | | |
|---|---|---|---|
| **95% C.I. difference** | **t-stat** | **p-value** | **Prop. dropped** |
| [0.00, 0.01] | 4.47 | < 0.01 | 0.23 |
| [0.01, 0.02] | 5.45 | < 0.01 | 0.45 |
| [0.02, 0.04] | 7.18 | < 0.01 | 0.68 |
| [0.03, 0.06] | 7.49 | < 0.01 | 0.90 |

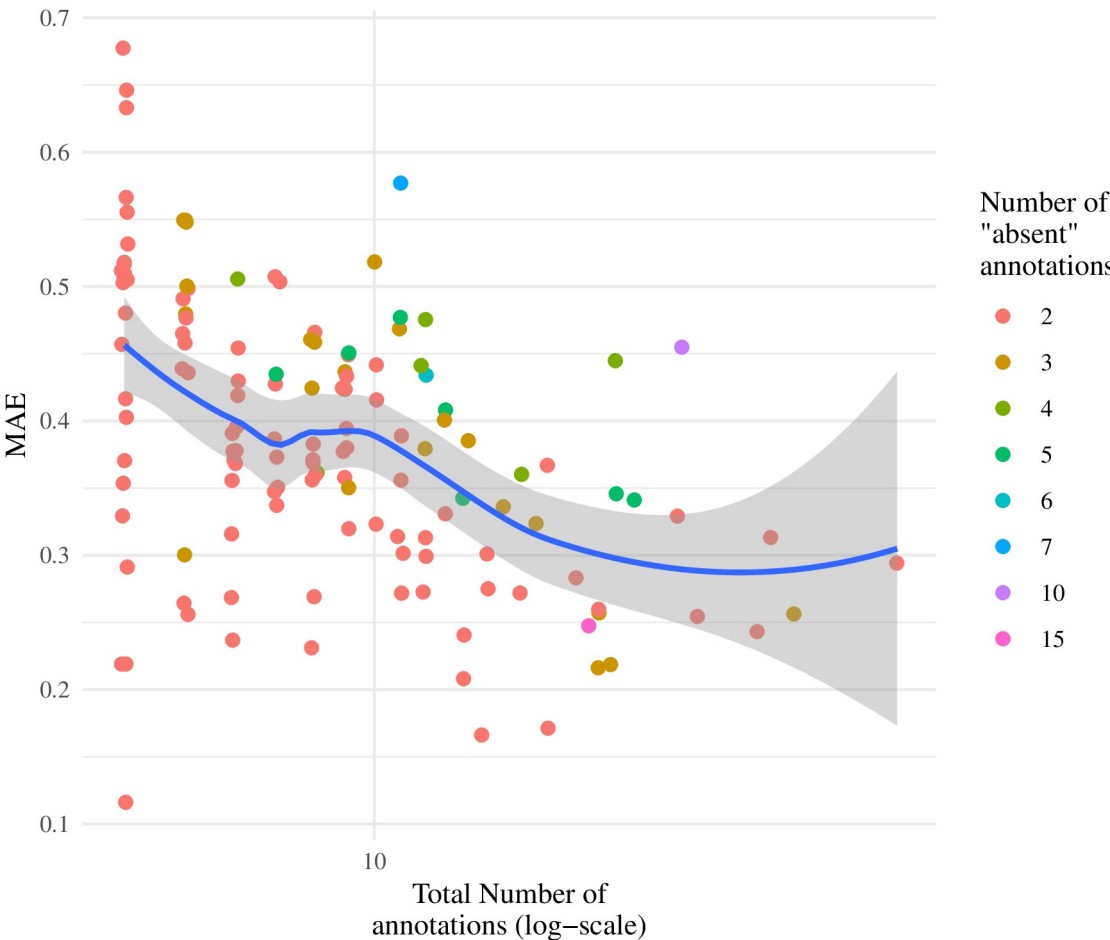

**Fig 5. Mean Absolute Error versus number of annotations on 138 trees.** Each point represents a single tree+function colored by the number of negative annotations ("absent"). The x-axis is in log-scale, and the figure includes a locally estimated scatterplot smoothing [LOESS] curve with a 95% confidence interval.

posterior distribution and then imputing function using each set of sampled parameter values. As before, we used a leave-one-out approach to make predictions, meaning that to predict any given leaf, we removed any annotation for that leaf and recalculated the probabilities described in Section 5.1.

In the low MAE example, Fig 6, which predicts the go term GO:0001730 on PANTHER family PTHR11258, we highlight three features of the figure. First, in box (a), the confidence intervals for the posterior probabilities in sections of the tree that have no annotations tend to be wider, which is a consequence of the degree to which posterior probabilities depend upon having nearby annotated nodes. Second, duplication events serve as turning points for function, as they can either trigger a functional loss, as highlighted in box (b), or a functional gain. The available data within the clade sets the overall trend for whether a function is present or not. The clade in box (b) is particularly interesting since there is only one duplication event and all of the annotated leaves have the function, which triggers a potential loss for the descendants of the duplication event. Finally, in regions in which labels are heterogeneous, it is harder to make a decisive prediction, as should be the case. This is why, in the leave-one-out predictions, the highlighted predictions in box (c) show wider confidence intervals and posterior probabilities closer to 0.5 than to the true state.

# PTHR11258

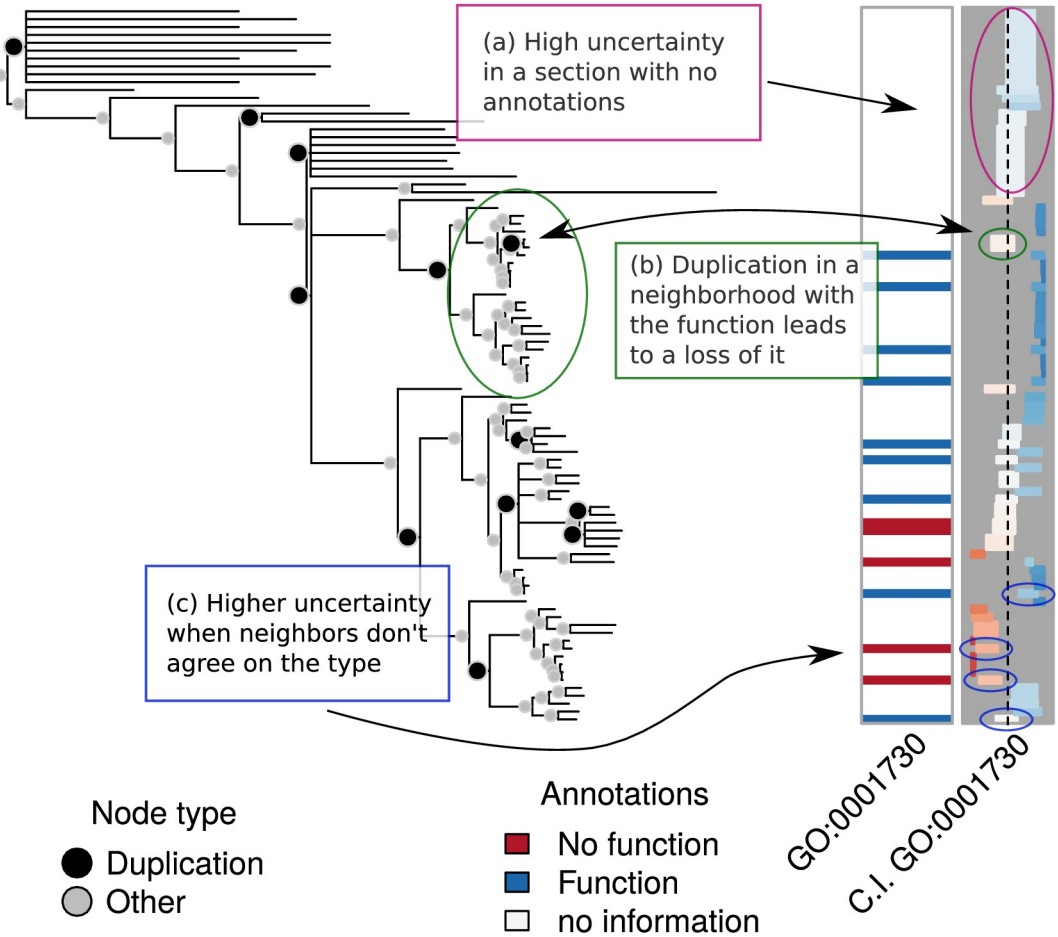

**Fig 6. Low MAE predictions.** The first set of annotations, first column, shows the experimental annotations of the term GO:0001730 for PTHR11258. The second column shows the 95% C.I. of the predicted annotations. The column ranges from 0 (left end) to 1 (right-end). Bars closer to the left are colored red to indicate that lack of function is suggested, while bars closer to the right are colored blue to indicate function is suggested. Depth of color corresponds to strength of inference. The AUC for this analysis is 0.91 and the Mean Absolute Error is 0.34.

Fig 7 illustrates one way in which high MAE can result. In particular, as highlighted by box (a), while the number of annotations is relatively large, (more than 10 over the entire tree, see Fig 3), the 'absent' (zero) annotations are distributed sparsely across the tree, having no immediate neighbor with an annotation of the same type. In fact, in every case, their closest annotated neighbors have the function. As a result, all zero-type annotations are wrongly labelled, with the model assigning a high probability of having the function. Similar to what is seen in the low prediction error case, duplication events within a clade with genes of mostly one type have a large impact on the posterior probabilities, as indicated by box (b).

In an effort to understand why the predictions were so poor in this case, we reviewed the family in detail. Because all GO annotations include references to the scientific publications upon which they are based, we were able to review the literature for genes in this family, the ER degradation enhancing mannosidase (EDEM) family. It turns out that the inconsistency of the

## PTHR45679

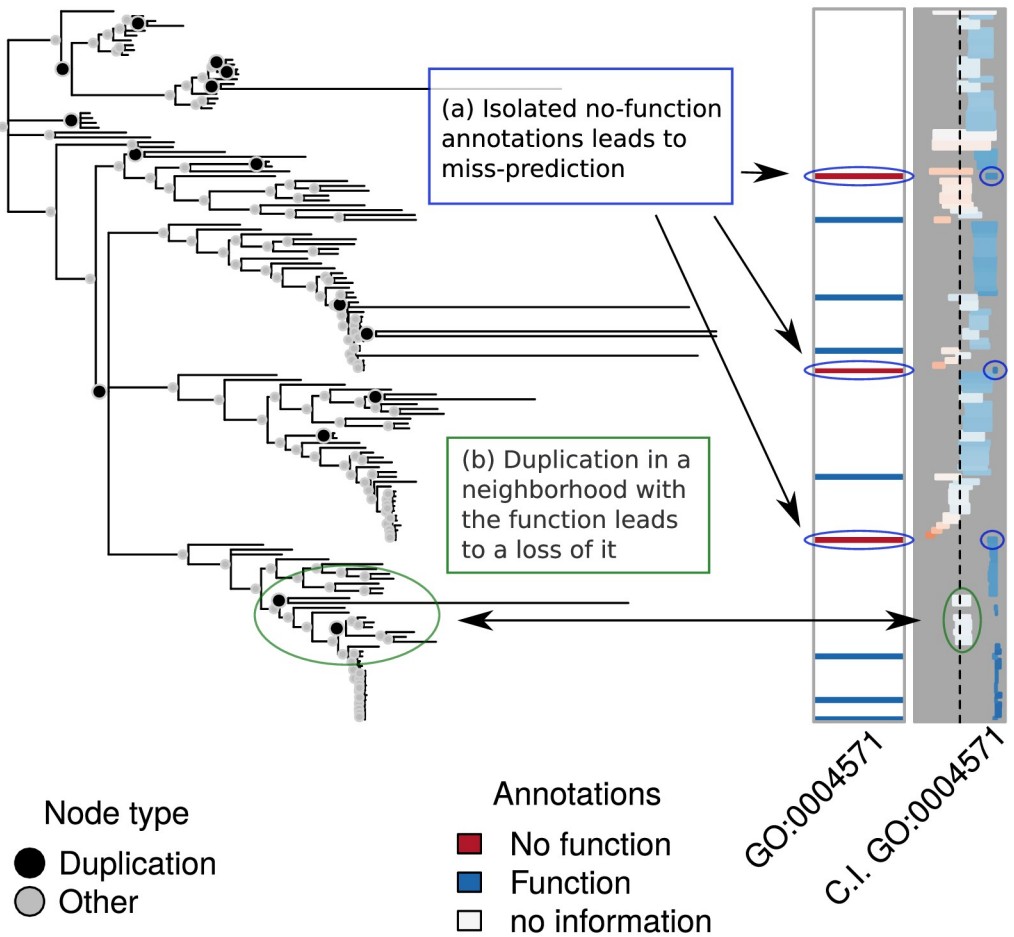

**Fig 7. High MAE predictions.** The first set of annotations, first column, shows the experimental annotations of the term GO:0004571 for PTHR45679. The second column shows the 95% C.I. of the predicted annotations using leave-one-out. Bars closer to the left are colored red to indicate that lack of function is suggested, while bars closer to the right are colored blue to indicate function is suggested. Depth of color corresponds to strength of inference. The AUC for this analysis is 0.33 and the Mean Absolute Error is 0.52.

annotations in the tree reflect a bona fide scientific controversy [25], with some publications reporting evidence that these proteins function as enzymes that cleave mannose linkages in gly-coproteins, while others report that they bind but do not cleave these linkages. This explains the divergent experimental annotations among members of this family, with some genes being annotated as having mannosidase activity (e.g. MNL1 in budding yeast), others as lacking this activity (e.g. mnl1 in fission yeast), and some with conflicting annotations (e.g. human EDEM1). The inability of our model to correctly predict the experimental annotations actually has a practical utility in this case, by identifying an area of inconsistency in the GO knowledgebase.

### 3.5 Comparison with SIFTER

Using our 1,1491 annotations over 1,325 genes, we ran SIFTER twice with leave-one-out cross-validation (LOO-CV) and truncation levels one and three, respectively, to build a

relevant benchmark for our model. We summarize the entire process we conducted in the following points:

1. We downloaded SIFTER version 2.1.1 from GitHub and installed it on a large server. We downloaded the GO tree and Pfam trees from SIFTER's website, which is dated 2015.

2. Using the EBI/EMBL API, we mapped our genes (proteins) to the corresponding Pfam trees available with SIFTER. In many cases, we obtained multiple hits per protein, which we kept. This means that for those proteins, we obtained more than one prediction.

3. Using an R wrapper written by ourselves, we ran SIFTER using our annotations together with SIFTER's reconciled Pfam trees using the options truncation level 1/3, so two runs, and cross-validation folds equal to zero, i.e., LOO-CV. In the case of the reminder parameters, we used SIFTER 2.1.1's default values.

4. Finally, looking at the intersection of the proteins annotated by both SIFTER and aphylo, we calculated MAE and AUC measurements for each method correspondingly.

In the cases in which we had more than one prediction per protein, we took the average. All code used to generate the benchmark, together with detailed instructions on how to setup SIFTER for such an analysis, and the data used to run SIFTER, is available on GitHub https://github.com/USCbiostats/sifter-aphylo. Even though we had more than 1,300 proteins with which to make the assessment, the final number of proteins that we were able to include in the analysis amounted to 147 proteins. Thus, the total number of annotations used in this part of the paper totaled 184, 18 of which were negative. Further details regarding the hardware used and SIFTER's installation can be found in Section 5.4.

For SIFTER, performing the entire LOO-CV process across 147 proteins using truncation level one, which is the fastest to run, took about 110 minutes, whereas, in the case of aphylo, the same took about one second. When using SIFTER with truncation level 3, a more detailed but slower mode, total analysis time was 200 minutes. All analyses used a single processing thread. Furthermore, calculating the posterior probabilities for all the leaves included in this analysis, which involves computing about 18,000 individual probabilities, took about 0.17 seconds total using aphylo.

With respect to prediction quality, as in previous sections we compare ground truth versus prediction at the annotation level. In particular, instead of looking at each set of annotations per protein and conducting a ROC-like analysis, as Engelhardt and others (2011) do, we leverage the fact that here we do have negative annotations, which allows us to perform a traditional ROC analysis. Fig 8 shows the ROC curves for both aphylo and SIFTER with truncation levels one (T = 1) and three (T = 3).

As shown in the figure, aphylo outperformed both SIFTER runs in terms of AUC, 0.72 versus 0.60 and 0.52. In the case of MAE, aphylo did a better job than SIFTER (T = 1) but it performed worse than SIFTER (T = 3). Overall, while SIFTER (T = 3) provides a greater deal of flexibility by allowing proteins to have at most three functions, since the data only has 18/184 negative annotations, 6 annotations that were classified as "absent" in SIFTER (T = 1) flipped to a positive probability of having the function, which is reflected in the poor AUC statistic; Section 5.4 discusses this issue further.

## 3.6 Discoveries

We now explore to what extent predictions from our model can be used to suggest function. Using the estimates from the pooled-data model with uniform priors, we calculated posterior probabilities for all the genes involved in the 138 GO trees+functions analyzed. Moreover, we

## Receiver Operating Characteristic

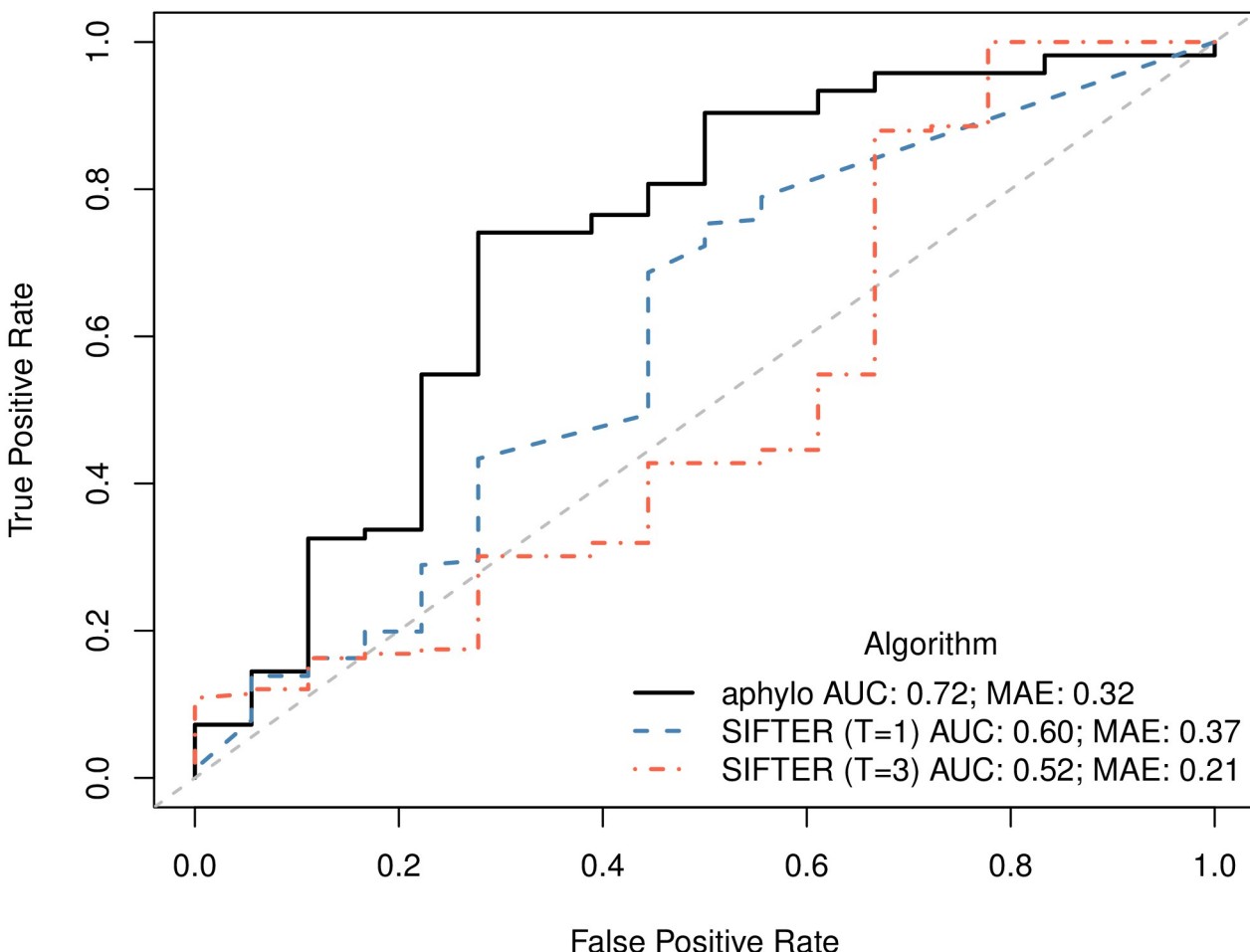

**Fig 8. ROC curve for aphylo and SIFTER predictions.** This figure includes 184 annotations on 147 proteins. Of the 184 annotations, 18 were negative. The corresponding AUC for these curves are 0.72 for aphylo, and 0.60 and 0.52 for SIFTER using truncation level 1 and truncation level 3 respectively.

calculated 95% credible intervals for each prediction using the posterior distribution of the parameters and from there obtained a set of potential new annotations.

While all posterior distributions are predictions, in order to focus on leaves for which state was predicted with a greater degree of certainty, we now consider the subset of predicted annotations whose 95% credible interval was either entirely below 0.1 or entirely above above 0.9, i.e., low or high chance of having the function respectively, which resulted in a list of 220 proposed annotations. The full list of predictions, including the programs used to generate it, is available in this website: https://github.com/USCbiostats/aphylo-simulations.

Using the QuickGO API, we made a search to see whether any of these proposed annotations were already present in the GO database. We compared our predictions to the annotations present in the 2020-10-09 release of the GO annotations (Fig 9). Of the 220 predictions, 46, eight of which we proposed to add with negative assertions, had no GO annotation to the proposed GO term, i.e., completely novel annotations. Two gene products had experimental annotations to the same terms we predicted (Hsd17b1 with estradiol 17-beta-dehydrogenase

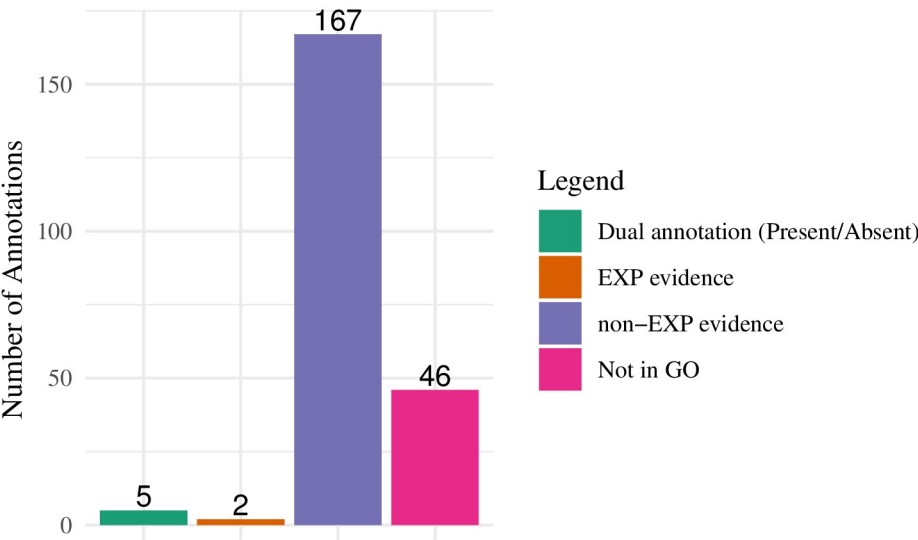

**Fig 9. Comparison of our 220 proposed annotations to annotations currently in the GO database.** Of the 220, 46 were not found in the GO database, 8 of which we proposed as negative assertions. Of the remaining 174 which we found in the GO database, five were inconsistent (having both a present and absent annotation), two had an experimental evidence code, and the remainder, 167, corresponded to annotations without experimental evidence.

activity (GO:0004303), and Abcb6 with mitochondrion (GO:0005739)); these annotations were made after the 2016-11-01 release we used for prediction. The remaining 172 predicted annotations matched only non-experimental annotations in the GO database, i.e. predictions from other methods. Five of these had both present and absent predictions by different methods, and the remaining 167 had no inconsistencies in annotation.

To gain further insight into our predictions, we manually analyzed all high-confidence predictions for genes in the mouse, shown in Table 7, a well-studied organism [26]. Of these 10 predictions, one matched a recently added experimental annotation in the GO database, estradiol dehydrogenase activity for Hsd17b1 [27]. We searched the literature, and found experimental evidence for two additional predictions: involvement of Bok in promoting apoptosis [28], and adenylate kinase activity for Ak2 [29]. Interestingly, all matching annotations in the

**Table 7. List of proposed annotations for mouse proteins.** The 95% CI corresponds to the posterior probability of the model. We suggest that values with the CI close to zero indicate absence of function and should be annotated with the qualifier "NOT". The Ref. column lists literature in which we found evidence of the corresponding annotation can be found, and the last column lists evidence codes found for annotations to the same term in the GO database (note that all corresponding GO database annotations are presence of function). *Notes*: (1) Exp. evidence found in literature. (2) Was not present in our training dataset, but we correctly predicted an experimentally supported annotation.

|  | Gene Symbol | 95% C.I. | Proposed Qualifier | Annotation | Ref. | Exp. Evidence |
|---|---|---|---|---|---|---|
| (1) | Bok | [0.93 0.98] |  | positive regulation of apoptotic process (GO:0043065) | [28] | ISO,IEA |
|  | Serpini1 | [0.93 0.99] |  | serine-type endopeptidase inhibitor activity (GO:0004867) |  | IBA,ISO,IEA |
|  | Aoc3 | [0.93 0.99] |  | primary amine oxidase activity (GO:0008131) |  | ISO,IEA,ISS |
| (1,2) | Hsd17b1 | [0.92 0.98] |  | estradiol 17-beta-dehydrogenase activity (GO:0004303) | [27] | IDA,ISO,IEA |
|  | Pnliprp1 | [0.02 0.08] | NOT | extracellular space (GO:0005615) |  | ISO |
|  | Serpinf2 | [0.92 0.98] |  | serine-type endopeptidase inhibitor activity (GO:0004867) |  | IBA,ISO,IEA |
|  | Ephb1 | [0.90 0.97] |  | protein tyrosine kinase activity (GO:0004713) |  | ISO,IEA |
|  | Abcb6 | [0.02 0.10] | NOT | mitochondrion (GO:0005739) |  | HDA,ISO,IEA |
| (1) | Ak2 | [0.90 0.97] |  | adenylate kinase activity (GO:0004017) | [29] | IBA,ISO,IEA |
|  | Ak3 | [0.02 0.10] | NOT | adenylate kinase activity (GO:0004017) |  | IBA,ISO,IEA |

GO database were assertions of presence of function, while three of our predictions were absence of the same function, so these are cases where our predictions are exactly counter to predictions from other methods. We confirmed that all three of our absence of function predictions are due to inference from an experimental absence annotation for the human ortholog of the corresponding mouse protein, so these predictions are not unreasonable given the input data.

## 4 Discussion

In this paper, we presented a model for the evolution of gene function that allows rapid inference of that function, along with the associated evolutionary parameters. Such a scheme allows for the hope of *automated* updating of gene function annotations as more experimental information is gathered. We note that our approach results in probabilistic inference of function, thereby capturing uncertainty in a concise and natural way. In simulation studies, using phylogenies from PANTHER, we demonstrate that our approach performs well. Furthermore, our computational implementation of this approach allows for rapid inference across thousands of trees in a short time period.

Regarding the question of when one should use aphylo versus SIFTER, or other methods, we note that if both negative and positive assertions are available, aphylo should perhaps be the method of choice since it explicitly uses both types of annotations which, as suggested in our analyses, should provide more accurate predictions than SIFTER. However, when only positive annotations are available, parameter estimation becomes problematic as estimates will tend to reflect a tree with positive assertions only. Unless one is willing to place strong priors on parameter values, parameter estimation under this scenario is not recommended. That said, for making predictions using parameters previously obtained from another training set (e.g. those in this paper), using aphylo is possible even when only positive annotations are available. Using pre-set parameters, the efficiency of aphylo would allow it to be incorporated into existing data-analysis pipelines to obtain a "second opinion" alongside other prediction methods, at a relatively little computational cost.

While the idea of using phylogenetic trees to inform species traits has been extensively developed in the phylogenetic regression literature [30, 31, 32, 33], one of several differences between these two approaches lies in how we use the trees. Closely related to spatial statistics, phylogenetic regression analyzes taxa-specific traits *while* controlling for "spatial correlation," informed by some distance metric based on the phylogeny. Instead, the approach presented here, models the evolutionary process itself. Furthermore, we believe that basing inference upon an evolutionary model allows us to capture biological properties of gene evolution, and that doing so results in more accurate inference. However, being based on a model also means being "wrong" [34]. We now discuss the ways in which we are wrong, and how we believe that despite this "wrongness" our approach is still useful.

*Sensitivity to misspecified trees.* Algorithms such as ours, and that of SIFTER, are built to exploit the information provided by the phylogenetic tree. Loosely speaking, leaves that are most closely related are most likely to have the same function. While it clearly makes sense to exploit this information, it does imply that errors in tree topology may lead to errors in inference of function. In principle, one could extend an algorithm such as ours to explicitly mix over the space of possible trees, but this would have several consequences:

1. We would require an evolutionary model for the topology of the tree itself, rather than just that for evolution of function. This would be a prerequisite for calculating tree-likelihoods, for example.

2. Computation time would increase by several orders of magnitude.

Given these concerns, and our desire to have an approach that retains computational tractability even for large phylogenetic trees (for which the space of possible tree topologies would be enormous), we have not explored this issue in this paper, but approaches in which one mixes over tree space as well have been tried in the coalescent literature for the evolution of individuals within a species, for example in the LAMARC 2.0 package [35].

*Branch length.* We have treated the probability of a change of function from node-to-node as unrelated to the length of the branch that connects those nodes. This choice follows the prevailing model of gene functional change occurring relatively rapidly after gene duplication, rather than gradually over time. Extension to models in which this is not true is conceptually straightforward, involving a move from discrete to continuous underlying Markov Chains. Parameters $\mu_{01}$ and $\mu_{10}$ are then treated as rates rather than probabilities: $\mathbb{P}(x_m = 1|x_{\mathbf{p}(m)} = 0, \tau), \quad \mathbb{P}(x_m = 0|x_{\mathbf{p}(m)} = 1, \tau)$, where $\tau$ is the length of the branch connecting that pair of nodes.

*Multiple gene functions or families.* When analyzing multiple gene functions, we have either treated them as if they all had the same parameter values, or were all completely independent. These cases represent two ends of a spectrum, and neither end is likely to be the truth. Even absent any specific knowledge regarding how genes might be related to each other, the sparseness of experimental annotation, and the low frequency of "*no function*" annotations, makes it desirable to take advantage of multiple annotations across functions in order to obtain better parameter inference (see Table 8). If there are $p$ gene functions in a particular gene family of interest, a fully saturated model would require $7 \times 2p$ parameters (4 sets of mutation rates, 2 sets of misclassification probabilities, and 1 set of root node probabilities). This, clearly, rapidly becomes very challenging, even for "not very large at all" $p$. Thus, more sophisticated approaches will be needed.

For now, we believe that the pooled analysis here demonstrates that there is likely to be some utility in developing approaches that can effectively impute annotations for multiple genes or gene functions jointly, particularly when, as is generally the case at present, annotation data is sparse. In future work we intend to explore hierarchical approaches to this problem. For example, it seems reasonable to assume that even if different gene families have different parameter values, the basic evolutionary biology would be similar across all gene families and this could be reflected by modeling the various parameters as random effects in a hierarchical framework. Thus, one would fit the entire ensemble of genes, or gene functions, simultaneously, estimating parameters for their overall means and variances across families.

**Table 8. Distribution of trees per number of functions (annotations) in PANTHER.** At least 50% of the trees used in this paper have 10 or more annotations per tree.

| N Ann. per Tree | N | Cum. count | Cum. prop. |
|---|---|---|---|
| 1 | 1173 | 1173 | 0.11 |
| 2 | 753 | 1926 | 0.17 |
| 3 | 675 | 2601 | 0.23 |
| 4 | 573 | 3174 | 0.29 |
| 5 | 567 | 3741 | 0.34 |
| 6 | 485 | 4226 | 0.38 |
| 7 | 504 | 4730 | 0.43 |
| 8 | 387 | 5117 | 0.46 |
| 9 | 323 | 5440 | 0.49 |
| 10 or more | 5666 | 11106 | 1.00 |

Such an approach would also allow for formal testing for parameter value differences between different families. While such an approach will be challenging to implement, we intend to pursue it in future work.

We also intend to explore approaches in which we use a simple loglinear model to capture the key features. For example, in such an approach we might include $p$ parameters for the marginal probabilities and $\binom{p}{2}$ parameters for pairwise associations for the mutation rates and baseline probabilities, for example, while treating the misclassification probabilities as independent. There are many possible extensions of this loglinear framework. For example, it seems likely that after a duplication event, as a function is lost in one branch a new function will be gained in that branch while the other branch remains unchanged. Such possibilities can be readily incorporated within this framework by modeling gain/loss probabilities jointly.

*Parameters suggest biological interpretations.* Our model, while simple, has several advantages and appears to perform well overall. We are able to determine parameters not only for one family at a time, but shared parameters over the set of all 138 protein families that have both positive and negative examples of a given function. The parameters have straightforward interpretations. In agreement with the prevailing model of the importance of gene duplication, the probability of function change (either gain or loss) derived from our model is much larger following gene duplication than following speciation. The high probability of an arbitrary function being present at the root of the tree ($\pi$) is consistent with the observation that functions are often highly conserved across long evolutionary time spans [36]. Our model also contains some features that may offer additional biological insights. Our sensitivity analysis shows that a small probability of functional loss following speciation is particularly important to prediction error. In other words, functions, once gained, strongly tend to be inherited in the absence of gene duplication. This includes not only molecular functions as generally accepted, but cellular components (the places where proteins are active), and biological processes (larger processes that proteins contribute to) as well.

*Utility of predictions.* The predictions from our method may have utility, both in guiding experimental work, or in highlighting areas of conflicting scientific results. High probability predictions are likely to be correct: for the ten such predictions we made for mouse genes, we found experimental evidence for six of them (which were not yet in the GO knowledgebase), and for the remaining four, we found no evidence of either presence or absence of that function. In our leave-one-out predictions of experimentally characterized functions of the EDEM family, we found cases where our predictions were particularly poor, but upon close examination turned out to be indicative of actual conflicts in experimental results from different studies. Deeper analyses of discrepant predictions could be helpful in identifying similar cases.

*Improving the input data using taxon constraints.* Another type information that can be leveraged is taxon constraints. Taxon constraints—which we define as a set of biological assumptions that restrict the set of possible values for given nodes on the tree (either by assuming gene function will be present or absent there)—can be used to inform our analysis. Within the context of our model, it is simple to specify that a clade can or cannot have a particular function, (essentially treating it as if it were fully annotated, without error). Thus, inclusion of such constraints would reduce the uncertainty in the model by effectively decreasing the overall depth of the unannotated parts of the tree (distance from the most recent common ancestor to the tip). More importantly, it would also act to increase the number of available annotations, most likely adding those of type *absent*, which as we show in Section 3, are the scarcest ones.

*Use in epidemiologic analyses.* We emphasize that the goal of this method is not simply to assign presence or absence of various gene functions to presently unannotated human genes, but to estimate the probabilities $\pi_{gp}$ that each gene $g$ has each function $p$. In analyzing

epidemiologic studies of gene-disease associations, we anticipate using this annotation information as "prior covariates" in a hierarchical model, in which the first level would be a conventional multiple logistic regression for the log relative risk coefficients $\beta_g$ for each gene $g$ and the second level would be a regression of these $\beta_g$ on the vector $\pi_g = (\pi_{pg})$ of estimated function probabilities. This second level model could take various forms, depending upon whether one thought the functions were informative about both the magnitude and sign of the relative risks or only their magnitudes. In former case, one might adopt a linear regression of the form $\beta_g \sim N(\alpha_0 + \pi'_g\alpha, \sigma^2)$. In the latter case, one might instead model their dispersion as $\beta_g \sim N(0, \lambda_g)$ or $\beta_g \sim \text{Laplace}(\lambda_g)$ where $\lambda_g = \exp\{\alpha_0 + \pi'_g\alpha\}$, corresponding to an individualized form of ridge or Lasso penalized regression respectively.

In summary, we have presented a parsimonious evolutionary model of gene function. While we intend to further develop this model to reflect additional biological features, we note that in its current form it has the following key features: (a) It is computationally scalable, making it trivial to jointly analyze hundreds of annotated trees in finite time. (b) It yields data-driven results that are aligned with our biological intuition, in particular, supporting the idea that functional changes are most likely to occur following gene-duplication events. (c) Notwithstanding its simplicity, it provides probabilistic predictions with an precision level comparable to that of other, more complex, phylo-based models. (d) Perhaps most importantly, it can be used to both support new annotations and to suggest areas in which existing annotations show inconsistencies that may indicate errors or controversies in those experimental annotations.

## 5 Material and methods

### 5.1 Prediction of annotations

Let $\tilde{D}_n^c$ denote an annotated tree with all tree structure and annotations below node $n$ removed, the complement of the induced sub-tree of $n$, $\tilde{D}_n$. Ultimately we are interested on the conditional probability of the $n$th node having the function of interest given $\tilde{D}$, the observed tree and annotations. Let $s \in \{0, 1\}$, then we need to compute:

$$
\begin{aligned}
\mathbb{P}(x_n = s | \tilde{D}) \quad &= \frac{\mathbb{P}(x_n = s, \tilde{D})}{\mathbb{P}(\tilde{D})} \\
&= \frac{\mathbb{P}(x_n = s, \tilde{D})}{\mathbb{P}(\tilde{D}|x_n = 1)\mathbb{P}(x_n = 1) + \mathbb{P}(\tilde{D}|x_n = 0)\mathbb{P}(x_n = 0)} \\
&= \frac{\mathbb{P}(\tilde{D}, x_n = s)}{\mathbb{P}(\tilde{D}, x_n = 1) + \mathbb{P}(\tilde{D}, x_n = 0)}
\end{aligned}
\tag{6}
$$

Using conditional independence (which follows from the Markovian property), the joint probability of $(\tilde{D}, x_n)$ can be decomposed into two pieces, the "pruning probability", which has already been calculated using the peeling algorithm described in section 2.2.2, and the joint probability of $(\tilde{D}_n^c, x_n)$:

$$
\begin{aligned}
\mathbb{P}(\tilde{D}, x_n = s) \quad &= \mathbb{P}(\tilde{D}|x_n = s)\mathbb{P}(x_n = s) \\
&\quad \text{(by conditional independence)} \\
&= \mathbb{P}(\tilde{D}_n|x_n = s)\mathbb{P}(\tilde{D}_n^c|x_n = s)\mathbb{P}(x_n = s) \\
&= \mathbb{P}(\tilde{D}_n|x_n = s)\mathbb{P}(\tilde{D}_n^c, x_n = s).
\end{aligned}
\tag{7}
$$

Using the law of total probability, the second term of (7) can be expressed in terms of $n$'s

parent state, $x_{\mathbf{p}(n)}$, as:

$$\mathbb{P}(\tilde{D}^c_n, x_n = s) \quad = \mathbb{P}(\tilde{D}^c_n, x_n = s | x_{\mathbf{p}(n)} = 1)\mathbb{P}(x_{\mathbf{p}(n)} = 1) +$$
$$\mathbb{P}(\tilde{D}^c_n, x_n = s | x_{\mathbf{p}(n)} = 0)\mathbb{P}(x_{\mathbf{p}(n)} = 0) \tag{8}$$

Again, given the state of the parent node, $x_{\mathbf{p}(n)}$, $x_n$ and $\tilde{D}^c_n$ are conditionally independent, and with $s' \in \{0, 1\}$, we have

$$\mathbb{P}(\tilde{D}^c_n, x_n = s | x_{\mathbf{p}(n)} = s') \quad = \mathbb{P}(\tilde{D}^c_n | x_{\mathbf{p}(n)} = s')\mathbb{P}(x_n = s | x_{\mathbf{p}(n)} = s')$$
$$= \frac{\mathbb{P}(\tilde{D}^c_n, x_{\mathbf{p}(n)} = s')}{\mathbb{P}(x_{\mathbf{p}(n)} = s')}\mathbb{P}(x_n = s | x_{\mathbf{p}(n)} = s')$$

A couple of observations from the previous equation. First, while $\tilde{D}^c_n$ includes node $\mathbf{p}(n)$, it does not include information about its state, $x_{\mathbf{p}(n)}$, since only leaf annotations are observed. Second, the equation now includes $\mathbb{P}(x_n | x_{\mathbf{p}(n)})$, this is, the model's transition probabilities, ($\mu_{01}$, $\mu_{10}$). With the above equation we can write (8) as:

$$\mathbb{P}(\tilde{D}^c_n, x_n = s) =$$
$$\mathbb{P}(\tilde{D}^c_n, x_{\mathbf{p}(n)} = 1)\mathbb{P}(x_n = s | x_{\mathbf{p}(n)} = 1) + \tag{8'}$$
$$\mathbb{P}(\tilde{D}^c_n, x_{\mathbf{p}(n)} = 0)\mathbb{P}(x_n = s | x_{\mathbf{p}(n)} = 0)$$

In (8′), the only missing piece is $\mathbb{P}(\tilde{D}^c_n, x_{\mathbf{p}(n)})$, which can be expressed as

$$\mathbb{P}(\tilde{D}^c_n | x_{\mathbf{p}(n)} = s')\mathbb{P}(x_{\mathbf{p}(n)} = s')$$

Furthermore, another application of the Markovian property allows us to decompose $\mathbb{P}(\tilde{D} | x_{\mathbf{p}(n)} = s')$ as a product of conditional probabilities. Thus, $\mathbb{P}(\tilde{D} | x_{\mathbf{p}(n)} = s')$ can be expressed as

$$= \mathbb{P}(\tilde{D}_{\mathbf{p}(n)} | x_{\mathbf{p}(n)} = s')\mathbb{P}(\tilde{D}^c_{\mathbf{p}(n)} | x_{\mathbf{p}(n)} = s').$$

(which, by definition, is)

$$= \left\{ \prod_{o \in \mathbf{Off}(\mathbf{p}(n))} \mathbb{P}(\tilde{D}_o | x_{\mathbf{p}(n)} = s') \right\} \mathbb{P}(\tilde{D}^c_{\mathbf{p}(n)} | x_{\mathbf{p}(n)} = s').$$

(and, taking the pruning probability of node $n$ out of the product operator, gives)

$$= \left\{ \prod_{o \in \mathbf{Off}(\mathbf{p}(n)) \setminus \{n\}} \mathbb{P}(\tilde{D}_o | x_{\mathbf{p}(n)} = s') \right\} \mathbb{P}(\tilde{D}^c_{\mathbf{p}(n)} | x_{\mathbf{p}(n)} = s') \times \mathbb{P}(\tilde{D}_n | x_{\mathbf{p}(n)} = s')$$
$$= \mathbb{P}(\tilde{D}^c_n | x_{\mathbf{p}(n)} = s')\mathbb{P}(\tilde{D}_n | x_{\mathbf{p}(n)} = s')$$

Where the last equality holds by definition of $\tilde{D}^c_n$. In essence, this shows that we can decompose the conditional probability $\tilde{D} | x_{\mathbf{p}(n)} = s'$ by splitting the tree at either $n$ or $\mathbf{p}(n)$. This allows us

to calculate $\mathbb{P}(\tilde{D}_n^c, x_{\mathbf{p}(n)})$:

$$
\begin{aligned}
\mathbb{P}(\tilde{D}_n^c, x_{\mathbf{p}(n)} = s') \quad &= \mathbb{P}(\tilde{D}_n^c | x_{\mathbf{p}(n)} = s')\mathbb{P}(x_{\mathbf{p}(n)} = s') \\
&\text{(multiplying and dividing by } \mathbb{P}(\tilde{D}_n | x_{\mathbf{p}(n)} = s') \text{ we get),} \\
&= \frac{\mathbb{P}(\tilde{D} | x_{\mathbf{p}(n)} = s')\mathbb{P}(x_{\mathbf{p}(n)} = s')}{\mathbb{P}(\tilde{D}_n | x_{\mathbf{p}(n)} = s')} \\
&\text{(Which, using the previous set of equations, is)} \\
&= \frac{\mathbb{P}(\tilde{D}, x_{\mathbf{p}(n)} = s')}{\mathbb{P}(\tilde{D}_n | x_{\mathbf{p}(n)} = s')}
\end{aligned}
\tag{9}
$$

The denominator of (9) can then be rewritten as follows:

$$
\begin{aligned}
\mathbb{P}(\tilde{D}_n | x_{\mathbf{p}(n)} = s') \quad &= \mathbb{P}(\tilde{D}_n | x_n = 1, x_{\mathbf{p}(n)} = s')\mathbb{P}(x_n = 1 | x_{\mathbf{p}(n)} = s') + \\
&\quad \mathbb{P}(\tilde{D}_n | x_n = 0, x_{\mathbf{p}(n)} = s')\mathbb{P}(x_n = 0 | x_{\mathbf{p}(n)} = s') \\
&\text{(given that we know that is } x_n, x_{\mathbf{p}(n)} \text{ is no longer informative for } \tilde{D}_n) \\
&= \mathbb{P}(\tilde{D}_n | x_n = 1)\mathbb{P}(x_n = 1 | x_{\mathbf{p}(n)} = s') + \\
&\quad \mathbb{P}(\tilde{D}_n | x_n = 0)\mathbb{P}(x_n = 0 | x_{\mathbf{p}(n)} = s').
\end{aligned}
$$

Now, we can substitute this into the denominator of (9) and write:

$$
\mathbb{P}(\tilde{D}_n^c, x_{\mathbf{p}(n)} = s') = \\
\frac{\mathbb{P}(\tilde{D}, x_{\mathbf{p}(n)} = s')}{\mathbb{P}(\tilde{D}_n | x_n = 1)\mathbb{P}(x_n = 1 | x_{\mathbf{p}(n)} = s') + \mathbb{P}(\tilde{D}_n | x_n = 0)\mathbb{P}(x_n = 0 | x_{\mathbf{p}(n)} = s')}.
\tag{9'}
$$

This way, the probability of observing $(\tilde{D}_n^c, x_n = s)$, (8'), equals:

$$
\begin{aligned}
\mathbb{P}(\tilde{D}_n^c, x_n = s) \quad &= \\
&\frac{\mathbb{P}(\tilde{D}, x_{\mathbf{p}(n)} = 1)\mathbb{P}(x_n = s | x_{\mathbf{p}(n)} = 1)}{\mathbb{P}(\tilde{D}_n | x_n = 1)(1 - \mu_{10}) + \mathbb{P}(\tilde{D}_n | x_n = 0)\mu_{10}} + \\
&\frac{\mathbb{P}(\tilde{D}, x_{\mathbf{p}(n)} = 0)\mathbb{P}(x_n = s | x_{\mathbf{p}(n)} = 0)}{\mathbb{P}(\tilde{D}_n | x_n = 1)\mu_{01} + \mathbb{P}(\tilde{D}_n | x_n = 0)(1 - \mu_{01})}
\end{aligned}
\tag{10}
$$

Together with the pruning probabilities calculated during the model fitting process, this equation can be computed using a recursive algorithm, in particular the pre-order traversal [37], in which we iterate through the nodes from root to leaves.

When node $n$ is the root node, the posterior probability can be calculated in a straightforward way:

$$
\begin{aligned}
\mathbb{P}(x_n = 1|\tilde{D}) &= \frac{\mathbb{P}(x_n = 1, \tilde{D})}{\mathbb{P}(x_n = 1, \tilde{D}) + \mathbb{P}(x_n = 0, \tilde{D})} \\
&= \frac{\mathbb{P}(\tilde{D}|x_n = 1)\mathbb{P}(x_n = 1)}{\mathbb{P}(\tilde{D}|x_n = 1)\mathbb{P}(x_n = 1) + \mathbb{P}(\tilde{D}|x_n = 0)\mathbb{P}(x_n = 0)} \\
&= \frac{\mathbb{P}(\tilde{D}_n|x_n = 1)\pi}{\mathbb{P}(\tilde{D}_n|x_n = 1)\pi + \mathbb{P}(\tilde{D}_n|x_n = 0)(1 - \pi)},
\end{aligned}
$$

since the terms $\mathbb{P}(\tilde{D}_n|x_n)$ have already been calculated as part of the pruning algorithm.

## 5.2 Monte carlo study

We assess model performance by quantifying the quality of our predictions under several scenarios using annotations constructed by simulating the evolutionary process on *real* trees obtained from PANTHER. In particular, we simulate the following scenarios:

1. **Fully annotated**: For each tree in PANTHER we simulated the evolution of gene function, and the annotation of that function at the tree tips, using the model described in this paper. For each tree we drew a different set of model parameters from the following Beta distribution:

| Parameter | $\alpha$ | $\beta$ | Mean $\alpha/(\alpha + \beta)$ |
|---|---|---|---|
| $\mu_{01d}$ | 38 | 2 | 0.95 |
| $\mu_{10d}$ | 10 | 10 | 0.50 |
| $\mu_{01s}, \mu_{10s}$ | 2 | 38 | 0.05 |
| $\pi$ | 2 | 38 | 0.05 |

We then used the simulated annotations to estimate the model parameters and gene function states. For this case we exclude mislabeling, i.e. all leaves are correctly annotated.

2. **Partially annotated**: Here, we took the set of simulations produced in scenario 1 above, but now estimated the model using a partially annotated tree. Specifically, we randomly dropped a proportion of leaf annotations $m \sim Uniform(0.1, 0.9)$. Once again, we assumed no mislabeling. (So, $\psi_{01} = 0$, $\psi_{10} = 0$).

3. **Partially annotated with mislabeling** Finally, we take the data from scenario 2 but allow for the possibility of mislabeling in the annotations. Specifically, for each tree we draw values for $\psi_{01}$ and $\psi_{10}$ from a Beta(2, 38) distribution.

In order to assess the effect of the prior distribution, in each scenario we performed estimation twice using two different priors: a well-specified prior, i.e., the one used during the data-generating-process, and a biased prior in which the $\alpha$ shape was twice of that of the data-generating-process.

The entire simulation study was conducted on USC's HPC cluster using the R package slurmR [38]. Data visualization was done using the R package ggplot2 [39].

**5.2.1 Prediction error.** Fig 10 shows the distribution of AUCs and Mean Absolute Errors [MAEs] for the third scenario by prior used and proportion of missing labels.

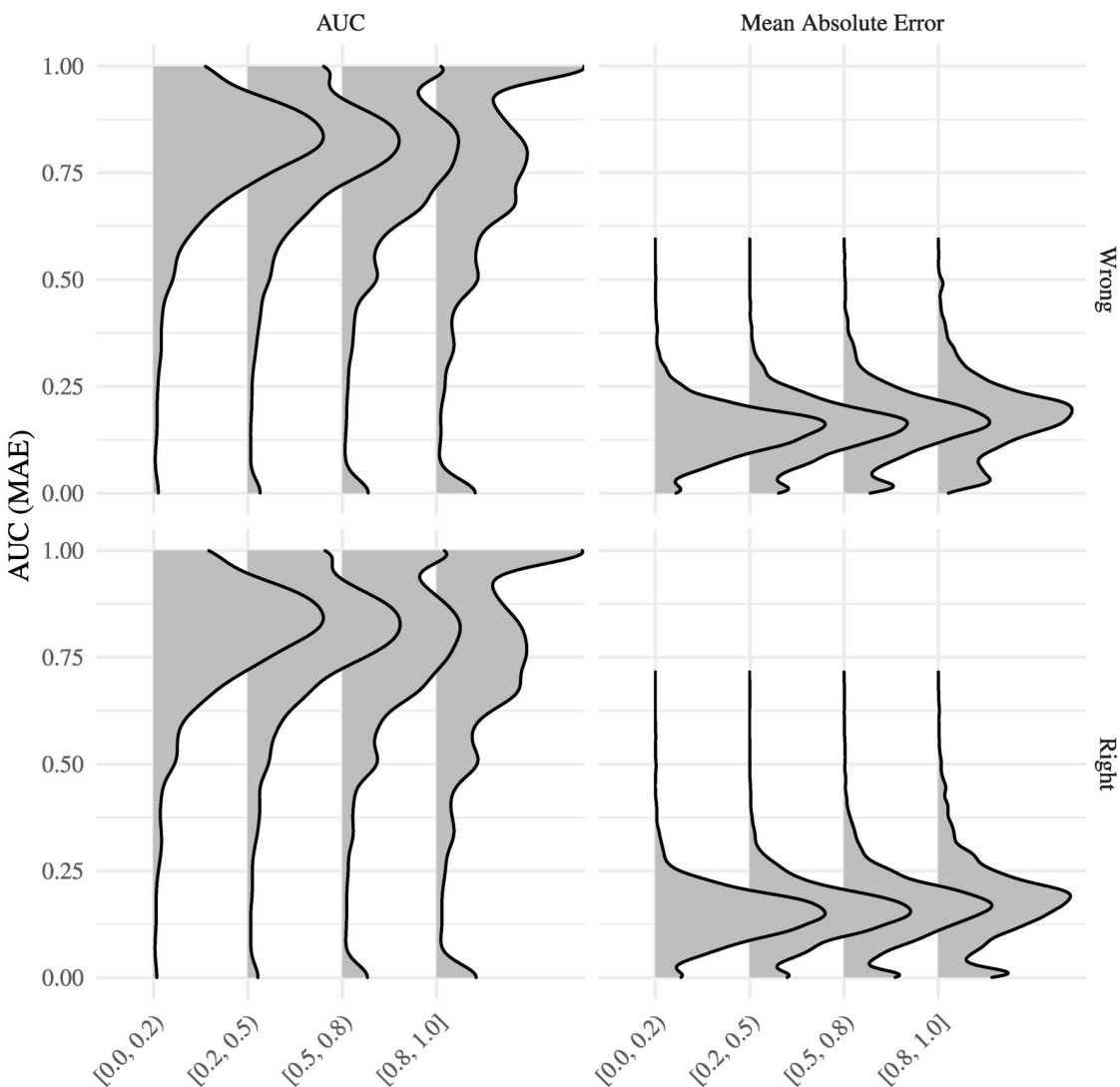

**Fig 10. Distribution of AUCs and MAEs for the scenario with partially annotated trees and mislabeling.** The x-axis shows the proportion of missing annotations, while the y-axis shows the score (AUC or MAE).

Overall, the predictions are good with a relatively low MAE/high AUC. Furthermore, as seen in Fig 10, both AUC and MAE worsen off as the data becomes more sparse (fewer annotated leaves).

**5.2.2 Bias.**   We now consider bias. Fig 11 shows the distribution of the empirical bias, defined as the population parameter minus the parameter estimate, for the first scenario (fully annotated tree). Since the tree is fully annotated and there is no mislabeling, the plot only shows the parameters for functional gain, loss and the root node probability. Of the three parameters, $\pi$ is the one which the model has the hardest time to recover, what's more, it generally seems to be over-estimated. On the other hand, $\mu_{01}$, $\mu_{10}$ estimates do significantly better than those for $\pi$, and moreover, in sufficiently large trees the model with the correct prior is able to recover the true parameter value.

Fig 12 shows the empirical distribution of the parameter estimates in the third simulation scheme: a partially annotated tree with mislabelling. As the proportion of missing annotations increases, the model tends to, as expected, do worse.

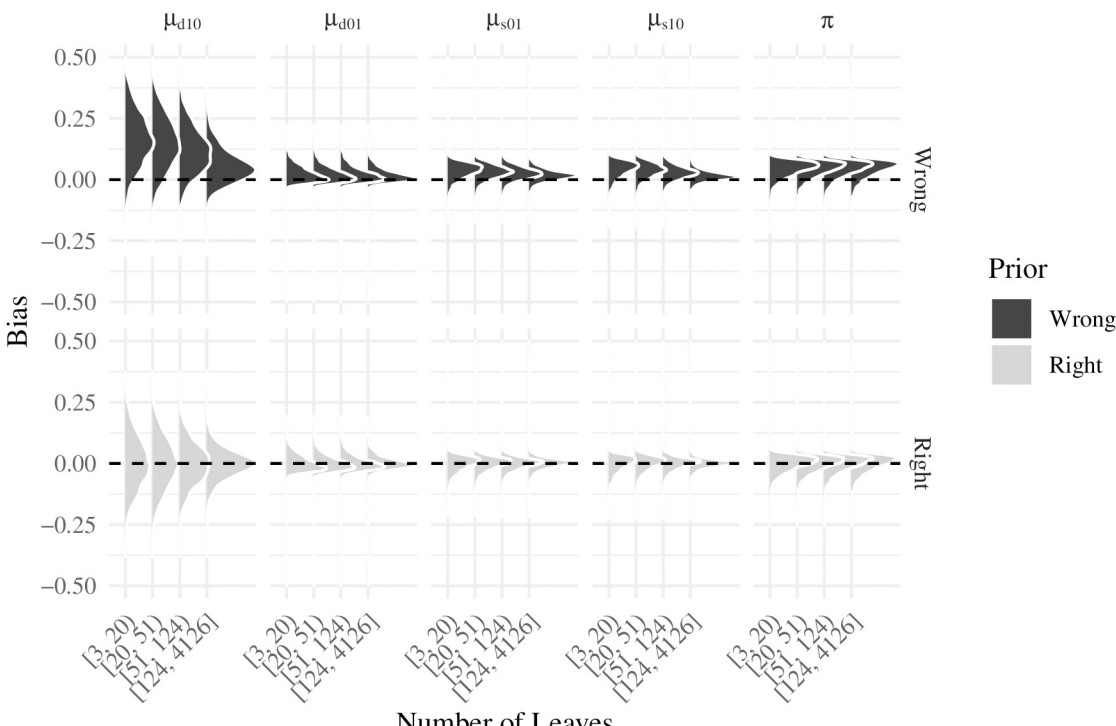

**Fig 11. Empirical bias distribution for the Fully annotated scenario by type of prior, parameter, and number of leaves.**

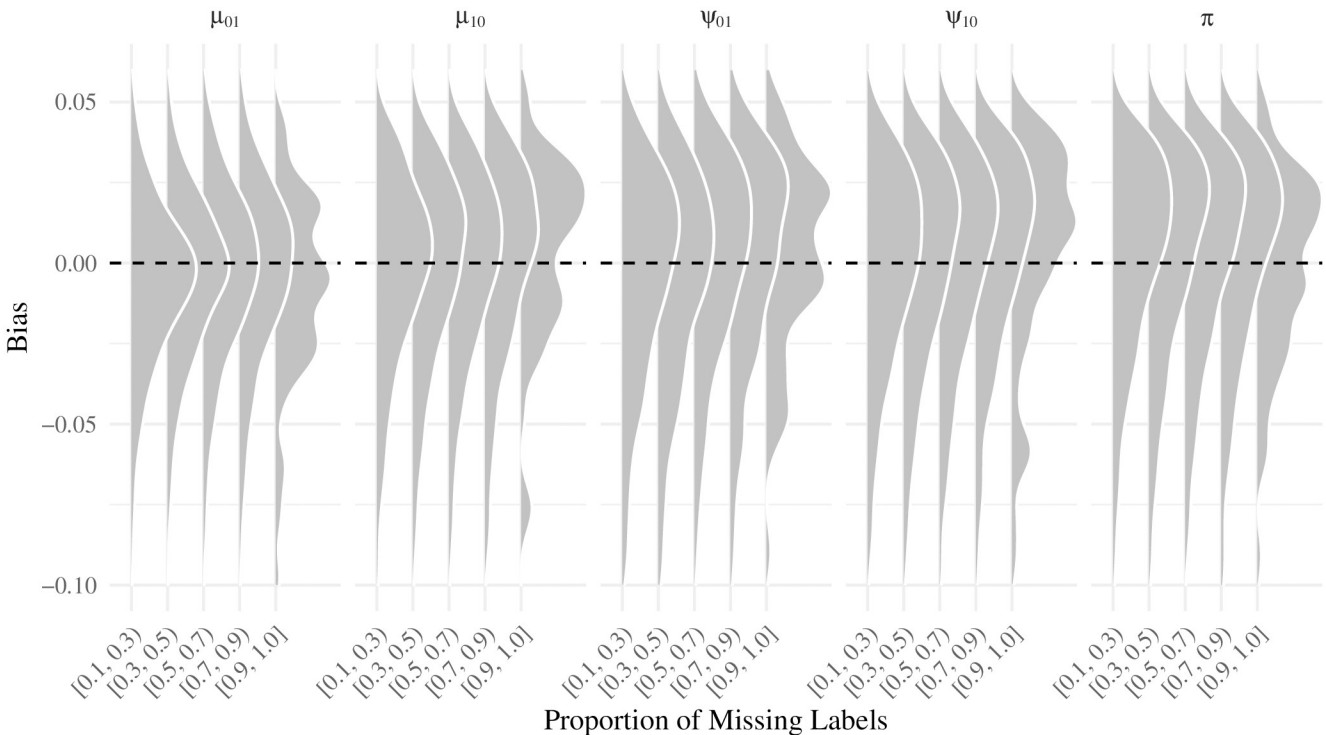

**Fig 12. Empirical bias distribution for the partially annotated scenario by parameter and proportion of missing labels.**

## 5.3 Limiting probabilities

In the case that the model includes a single set of gain and loss probabilities, (i.e. no difference according to type of node), in the limit as the number of branches between the root and the node goes to infinity, the probability that a given node has a function can be calculated as

$$
\begin{aligned}
\mathbb{P}(z_n = 1) = \quad & (1 - \mu_{10})\mathbb{P}(z_{n-1} = 1) + \mu_{01}\mathbb{P}(z_{n-1} = 0), \\
& \text{since in the limit } \mathbb{P}(z_n = 1) = \mathbb{P}(z_{n+k} = 1), \quad \forall k \\
= \quad & (1 - \mu_{10})\mathbb{P}(z_n = 1) + \mu_{01}\mathbb{P}(z_n = 0), \\
& \text{finally} \\
\mathbb{P}(z_n = 1) = \quad & \frac{\mu_{01}}{\mu_{01} + \mu_{10}}
\end{aligned} \tag{11}
$$

However, when the gain and loss probabilities differ by type of node, the unconditional probability of observing having function is computed as follows:

$$
\mathbb{P}(z_n = 1) = \mathbb{P}(z_n = 1 | C_n = D)\mathbb{P}(C_n = D) + \mathbb{P}(z_n = 1 | C_n = \neg D)\mathbb{P}(C_n = \neg D) \tag{12}
$$

Where $C_n \in \{D, \neg D\}$ denotes the type of event (duplication or not duplication, respectively). We need to calculate $\mathbb{P}(z_n = 1 | C_n = D)$ and $\mathbb{P}(z_n = 1 | C_n = \neg D)$. WLOG let's start by the first:

$$
\begin{aligned}
\mathbb{P}(z_n = 1 | C_n = D) \quad = \quad & \mathbb{P}(z_n = 1 | C_n = D, z_{n-1} = 1)\mathbb{P}(z_{n-1} = 1 | C_n = D) + \\
& \mathbb{P}(z_n = 1 | C_n = D, z_{n-1} = 0)\mathbb{P}(z_{n-1} = 0 | C_n = D) \\
= \quad & (1 - \mu_{10}{}^D)\mathbb{P}(z_{n-1} = 1 | C_n = D) + \mu_{01}{}^D \mathbb{P}(z_{n-1} = 0 | C_n = D) \\
& \text{Now, following the same argument made in (11),} \\
\mu^D = \quad & \frac{\mu_{01}{}^D}{\mu_{01}{}^D + \mu_{10}{}^D}
\end{aligned}
$$

Where $\mu^D = \mathbb{P}(z_n | D_n, D_{n-1})$. Likewise, $\mu^{\neg D} = \frac{\mu_{01}{}^{\neg D}}{\mu_{01}{}^{\neg D} + \mu_{10}{}^{\neg D}}$. Observe that the parameter is only indexed by the class of the $n$-th leaf as the class of its parent is not relevant for this calculation.

$$
\mathbb{P}(z_n = 1) = \mu^D \mathbb{P}(C_n = D) + \mu^{\neg D}\mathbb{P}(C_n = \neg D) \tag{12$'$}
$$

Where the probability of a node having a given class can be approximated by the observed proportion of that class in the given tree.

## 5.4 SIFTER analysis

For the analysis using GO annotations and their corresponding PANTHER trees, we focused on genes for which there were at least two with an annotation of the type "present" (i.e. a "1") and two others with an annotation of type "absent" (i.e. a "0") for the same function within the same tree, thus at least four annotations. These requirements yielded the initial number of ~1, 500 annotations over ~1, 300 proteins which we used for fitting our evolutionary model. But when mapping those proteins to Pfam trees, we were not able to use all 1,300 with SIFTER, as many of the resulting trees either ended up with a single protein, which precludes SIFTER from running the LOO-CV algorithm, or with a single function, which means that SIFTER cannot be used, since it requires at least two functions for inference; if there is just one function, it simply infers all leaves as having the function. Because of these considerations, the initial set of candidate proteins that SIFTER could analyze reduced from 1,300 to 472.

After running SIFTER with the 472 proteins, the total number of proteins that could be compared with our method was further reduced to 147. The reason for this reflect on a couple of issues that we observed during the estimation process:

1. In a pre-processing step, SIFTER prunes the GO tree, which ultimately translates into dropping some annotations. For example, if we passed SIFTER a dataset with 2 GO terms, in some cases it would drop one of those and proceed with the calculations using only a single GO term. This is described in detail in page 18 of the Supplemental Materials of Engelheardt et al. (2011), "In the GO DAG, we first prune all ancestors of nodes with annotations (even if the ancestors themselves have annotations), then we prune all non-annotated nodes. This leaves a set of candidate functions that are neither ancestors nor descendants of each other, ensuring there are no deterministic dependencies between them in terms of the semantic network."

2. In some other cases, the Pfam version of SIFTER, which is dated 2015, did not include some of the families we identified with the EBI/EMBL API.

3. In another handful of cases, SIFTER was not able to find some of the proteins included in the analysis. Again, similar to the previous point, this was mostly due to the fact that the Pfam database is outdated.

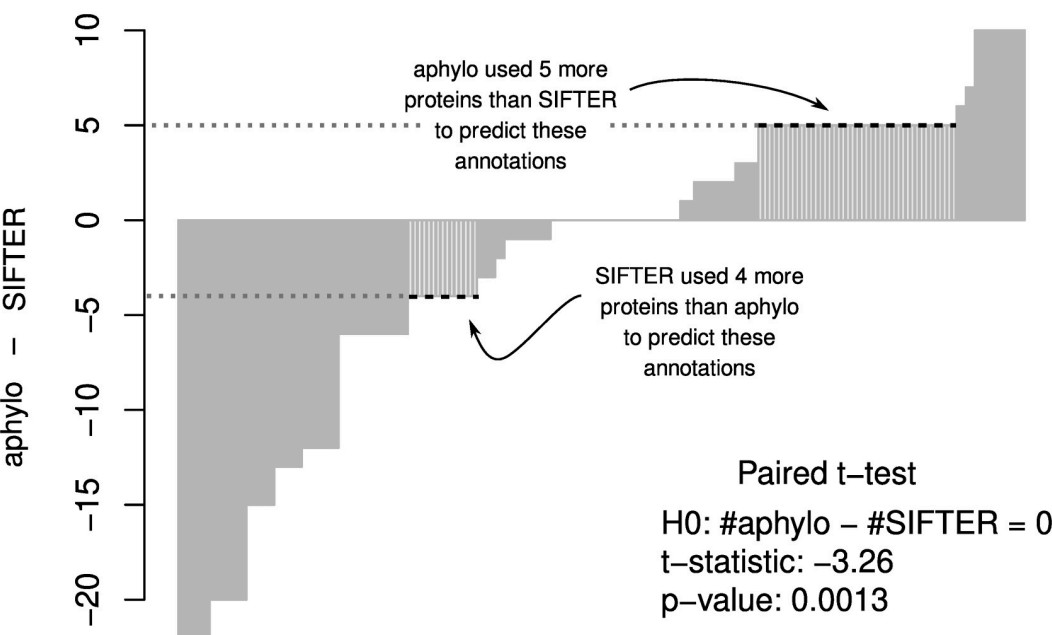

**Fig 13. Difference in the number of input proteins used for predictions.** Each bar represents a single annotation (prediction) to be made. The y-axis shows the difference between the number of input proteins used by aphylo, and SIFTER. Negative values indicate that SIFTER included more proteins as input for making the prediction, whereas positive values indicate that aphylo included more proteins as input. A paired t-test shows that, on average, SIFTER included more proteins than aphylo for each one of its calculations.

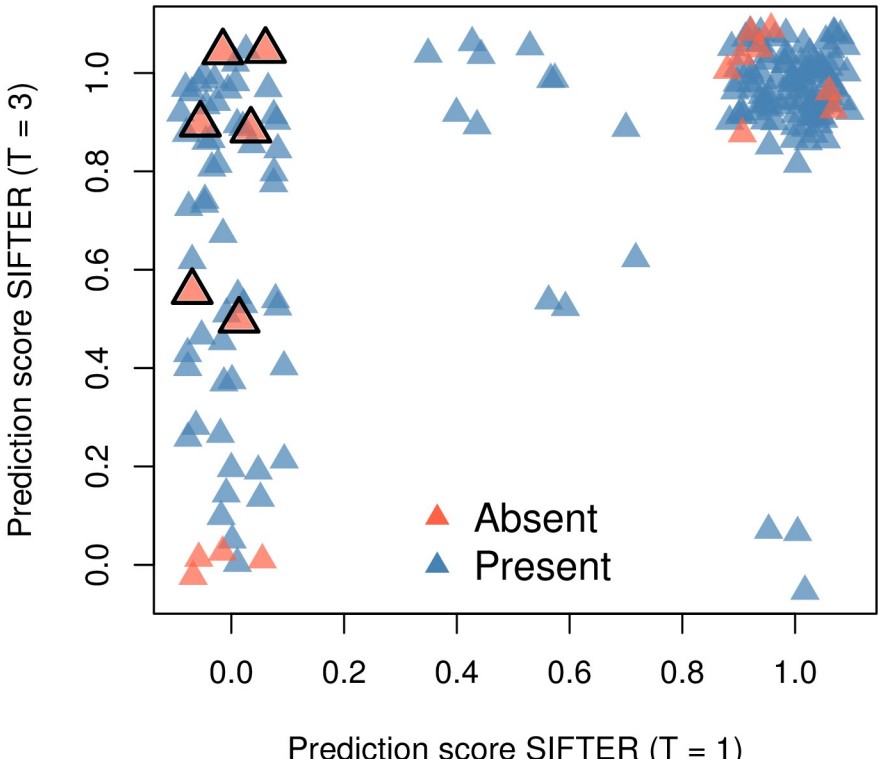

**Fig 14. Scatter plot comparing all 184 prediction scores between SIFTER with truncation level one, and SIFTER with truncation level 3.** Red triangles correspond to negative annotations, while blue triangles mark positive annotations. The six negative annotations with a black border highlight observations that were correctly classified with truncation one, but were misclassified using truncation level 3. The coordinates were jittered to avoid overlapping.

4. After these cases were excluded, there was only one case in which SIFTER failed to converge after one hour of analysis and another case in which the final set of proteins was of size one, so LOO-CV was not possible.

Since both methods ended up using different sets of trees, we assessed whether either of the two used more or less proteins as input for making their predictions, which is shown in Fig 13. We can see that on average SIFTER had more proteins available than aphylo to compute the posterior probabilities. Moreover, a paired t-test showed that that difference is significant, with SIFTER using an average of about three more proteins than aphylo per prediction. This difference is not surprising as SIFTER was designed to make inferences on sets of GO annotations, which ultimately translates into jointly analyzing proteins with disjoint sets of annotations.

Another interesting point to highlight is the difference in SIFTER's prediction scores using truncation level one [ST1] versus truncation level three [ST3], which is shown in Fig 8. Oddly enough, while SIFTER with truncation level 3 shows a smaller MAE (better prediction), the AUC is worse. Fig 14 shows a jittered scatter plot of each truncation level's prediction scores. Looking at that figure, we can see that positive annotations that were classified as negative with ST1 now have a positive probability under ST3. Since most of the annotations in the data are positive annotations, this reduces the mean square error. At the same time, six proteins that were correctly marked as not having the function in ST1, under ST3 have now a positive probability of having the function. Since the data have only 18 negative annotations, this has a great impact on the calculated AUCs.

Finally, it is worthwhile noting that the entire installation of SIFTER required approximately 150 Gb. While SIFTER is well documented, the python wrappers— SIFTER's current main user interface—could not be used to run LOO-CV, which is why we opted to use the core program, written in Java. This also restricted the set of annotations available to use with SIFTER.

## 5.5 Computational resources

The Monte Carlo simulation study featured in Section 5.2 was conducted on USC's Center for High Performance Computing Cluster, a Slurm cluster. In general, simulations and annotations were performed using between 200 up to 400 threads.

The analyses using GO annotations, involving the 1,300 experimentally annotated proteins were run on a large CentOS 7 server with 20 physical cores Intel(R) Xeon(R) CPU E5-2640 v4 @ 2.40GHz and 250 Gb of memory. Nevertheless, all calculations involving the 1,300 proteins were done using a single computational thread and less than one GB of memory.

## Author Contributions

**Conceptualization:** Duncan C. Thomas, Paul D. Thomas.

**Data curation:** Huaiyu Mi.

**Formal analysis:** George G. Vega Yon, John Morrison.

**Funding acquisition:** Duncan C. Thomas, Huaiyu Mi, Paul D. Thomas, Paul Marjoram.

**Investigation:** George G. Vega Yon.

**Methodology:** George G. Vega Yon, Duncan C. Thomas, Paul D. Thomas, Paul Marjoram.

**Resources:** Duncan C. Thomas, Huaiyu Mi, Paul D. Thomas, Paul Marjoram.

**Software:** George G. Vega Yon, Duncan C. Thomas.

**Supervision:** Duncan C. Thomas, Paul D. Thomas, Paul Marjoram.

**Validation:** George G. Vega Yon, John Morrison, Huaiyu Mi, Paul D. Thomas.

**Visualization:** George G. Vega Yon.

**Writing – original draft:** George G. Vega Yon, Paul D. Thomas.

**Writing – review & editing:** George G. Vega Yon, Duncan C. Thomas, Paul D. Thomas, Paul Marjoram.

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
