## [Decision Letter · Decision Letter 0]

26 Jul 2020

Dear Vega Yon,

Thank you very much for submitting your manuscript "On the automatic annotation of gene functions using observational data and phylogenetic trees" for consideration at PLOS Computational Biology.

As with all papers reviewed by the journal, your manuscript was reviewed by members of the editorial board and by several independent reviewers. In light of the reviews (below this email), we would like to invite the resubmission of a significantly-revised version that takes into account the reviewers' comments.

All three reviewers raise serious concerns regarding a lack of comparisons to existing methods and/or baselines, which need to be specifically addressed in the revised version.

We cannot make any decision about publication until we have seen the revised manuscript and your response to the reviewers' comments. Your revised manuscript is also likely to be sent to reviewers for further evaluation.

Sincerely,

Mohammed El-Kebir

Guest Editor

PLOS Computational Biology

Jian Ma

Deputy Editor

PLOS Computational Biology

All three reviewers raise serious concerns regarding a lack of comparisons to existing methods and/or baselines, which need to be addressed.

Reviewer's Responses to Questions

**Comments to the Authors:**

**Reviewer #1: **The authors present a computational method for modeling the evolution of gene/protein function annotations in a phylogenetic tree. Since such annotations are typically sparse and noisy, computational approaches for inferring annotations can aid biological and biomedical analyses, and the authors' use of phylogenetic trees for their evolutionary model provides biological justification for the inferred gene functions. The paper is generally clear and well written with appropriate mathematical notation for describing their method, which is easy to understand. However, it would benefit by addressing several questions and concerns.

1. How sensitive is the method to the topology of the phylogenetic trees or missing or spurious edges in the phylogenetic trees?

2. What is the computational complexity of the method as a function of standard properties (number of nodes, number of edges, depth, etc.) of the phylogenetic tree? The method seems to be very efficient in practice, but it is not clear how well it scales.

3. The authors could be more rigorous/precise about how they evaluate their method's performance. Some statements can be difficult to understand, such as "about 73% accuracy using the area under the curve statistic". Some mathematical and statistical terms are used inconsistency or interchangeably, such as accuracy and AUC/AURC and MAE, or are described in non-standard ways. Unfortunately, these issues may be distracting and confusing to some readers.

4. The method appears to be very robust to missing data and surprisingly seems to result in higher performance with higher missingness. The MAEs of their results also appear to be close to 0.5 for many of the experiments, where 0.5 is what one would expect for a random classifier. It is not clear how well the method is performing or that it is performing as one might expect, e.g., better performance with more data. (Also, the authors say that a "paired t-test found a significant drop in the accuracy levels", but they're not comparing accuracy, the effect size seems very small, and it is unclear if the comparison is overpowered.)

5. Similarly, how does this method compare to similar approaches that use evolutionary models on phylogenetic trees? How does this method compare to other approaches that do not use phylogenetic trees? How does it compare to a very simple baseline approach? Do they have similar MAEs? The authors say that many methods have been developed for inferring gene function, so it would be helpful to compare with some of them. The authors mention SIFTER, but they only compare their method's run times with the projected run times from SIFTER. They do not compare the results or actual run times.

6. The authors helpfully perform a detailed sensitivity analysis of their model's parameters. They also apply their method to a number of GO terms and phylogenies from the PANTHER database, but they limit themselves to a small subset (141 of 15,000) of the phylogenies that are relatively well annotated and therefore potentially unrepresentative of what many users may encounter. There could be more discussion of the biological results.

**Reviewer #2:** Summary:

This paper presents a new phylogeny-based method, *aphylo*, for predicting gene ontology (GO) annotations from data.

The data are gene trees, where each leaf represents a protein sequence that has been annotated based on function (1 = function present and 0 = function absent). Importantly, each gene tree has been **reconciled** based on a rooted species tree, so that the gene tree is rooted and its internal nodes are labeled by evolutionary events (e.g. speciation or duplication). The training data are reconciled gene trees where all leaves have annotations, and the testing data are reconciled gene trees where some but not all leaves have annotations. The goal is then to predict the missing annotations. The proposed method, *aphylo*, achieves this by defining a generic model of function evolution and then utilizing this model within a Bayesian framework.

At a high level, function is modeled as having evolved down the gene tree, meaning that function occurs at the root with some probability and then can be gained or lost on each edge with some probability that depends on whether the node above the edge represents a speciation or duplication event. The posterior distributions of the model parameters (e.g., the probability of gaining function after a gene duplication event) are estimated from the training data within a Bayesian framework (although MLE and MAP estimation is also enabled). Then, at each node, conditioned on these distributions, the posterior probability of function can be computed --- this is used to predict whether function is present (1) or absent (0) at leaves without annotations.

The major weakness of this paper is that *aphylo* (which seems simpler than other phylogeny-based methods for gene ontology annotation) is not benchmarked against other methods (major comment #1).

Because this paper does not evaluate whether *aphylo* improves upon the accuracy or speed of other methods for gene ontology annotation (e.g. those evaluated in the CAFA2 study), it is difficult to evaluate its significance. Therefore, I recommend "Reject".

Major Comments:

1. On line 509, the paper reads: "Notwithstanding [*alphylo*'s] simplicity, it provides probabilistic predictions with an accuracy level comparable to that of other more complex, phyla-based models." This is statement is surprising given that the proposed method, *aphylo*, was not benchmarked against other methods (at least I didn't notice this in the main paper). In lines 253-270, there is a brief comparison of *aphylo* to SIFTER [1,2]; however, this discussion is focused on running time not prediction (NOTE: Figures 1-8 in the main text do not seem to show comparisons to SIFTER). Although SIFTER may be too computationally intensive to run on all datasets evaluated in this study, comparisons could be made on at least a small subset of the datasets. If it is impossible to make a comparison to SIFTER, then a comparison could be made to a "base-line method", for example the BLAST-based approach used in the "term-centric evaluation" from the second Critical Assessment of Functional Annotation study (CAFA2) [3]. Term-centric experiments evaluate methods for binary classification tasks, where given a protein sequence, an ontology term (related to molecular function, biological process, cellular component, or human phenotype) is assigned (1 = present and 0 = absent). Several different methods, including SIFTER 2.4 [1,2], were evaluated in CAFA2. Interestingly, Jones-UCL [3] outperformed SIFTER (Fig 7 in [3]); therefore, it seems relevant to compare *aphylo* to Jones-UCL, SIFTER, and a base-line method (assuming that it is computationally feasible to perform such experiments).

2. Figure 1 shows the results of several experiments on simulated datasets to evaluate *aphylo* under different conditions (NOTE: Figure 1 would benefit from having the 8 subplots labeled "A", "B", ... "H"). Under ideal conditions (white boxes), the mean MAE is approximately 0.38 (MAE of 0.50 is no better than a random coin toss); however, the MAE takes on a HUGE range. It would be beneficial to do a more fine-grained analysis of datasets with very low and very high MAE. Although justification for showing MAE (instead of ROC curves) is provided, my opinion is that showing ROC curves in addition would be helpful for many readers.

3. On line 320, the paper reads: "Finally, we evaluated the impact of removing annotations on accuracy. While a significant drop may not be obvious from the last box in Figure 1, a paired t-test found a significant drop in the accuracy levels." From the description starting on line 536, it sounds like annotations were deleted from gene trees uniformly at random. Does this pattern of missing annotations reflect the pattern of missing annotations in the biological datasets (especially for the lower rates of missing data)? Given the feature example where co-location of missing annotations impacts accuracy, it may be useful to evaluate this issue further (major comment #2).

4. Gene duplication has a huge impact on the probability of gaining function (Tables 3 and 4). Because the gene tree and its reconciliation are estimated from molecular sequence data, it seems important to consider how the accuracy of the estimated gene trees and their reconciliations may impact the accuracy of *aphylo*. Note that gene trees reconciliation may be especially challenging when there is hidden paralogy (https://ecoevorxiv.org/wzcbg/). It would be interesting if strange results, such as parallel evolution of function, ormany gains/losses in function, could be explained gene tree estimation error or reconciliation error. Along these lines, it also would be helpful to briefly note how gene trees were estimated and reconciled for the PANTHER v11 (as well as if branch support was estimated).

Minor Comments:

1. Section 3.5 (Discoveries) seems quite valuable, and it may be worthwhile to expand this section and to display the key findings (e.g. for the 10 mouse genes) in a table.

2. This paper models the evolution of a binary trait (in this case the presence or absence of a particular function), so it may be beneficial to discuss the connections to related work in this area (e.g., [5, 6, 7, 8]).

References:

1. B.E. Engelhardt, M.I. Jordan, J.R. Srouji, S.E. Brenner. Genome-scale phylogenetic function annotation of large and diverse protein families. Genome Research 21:1969-1980, 2011. doi: 10.1101/gr.104687.109

2. S.M.E. Sahraeian, K.R. Luo, S.E. Brenner. SIFTER search: a web server for accurate phylogeny-based protein function prediction. Nucleic Acids Research 43:W141-W147, 2015. doi: 10.1093/nar/gkv461

3. Y. Jiang, T. Oron, et al. An expanded evaluation of protein function prediction methods shows an improvement in accuracy. Genome Biology 17:184, 2016. doi: 10.1186/s13059-016-1037-6

4. D. Cozzetto, D. W. Buchan, K. Bryson, and D. T. Jones. Protein function prediction by massive integration of evolutionary analyses and multiple data sources. BMC Bioinformatics 14(Suppl 3):S1, 2013. doi: 10.1186/1471-2105-14-S3-S1

5. A.R. Ives and T. Garland, Jr. Phylogenetic logistic regression for binary dependent variables. Systematic Biology 59:9-26, 2010. doi: 10.1093/sysbio/syp074.

6. L.T. Ho and C. Ane. A Linear-Time Algorithm for Gaussian and Non-Gaussian Trait Evolution Models. Systematic Biology 63:397-408, 2014. doi: 10.1093/sysbio/syu005

Also see https://rdrr.io/cran/phylolm/

7. L.J. Revell. Two new graphical methods for mapping trait evolution on phylogenies. Methods in Ecology and Evolution 4:8, 2013. doi: 10.1111/2041-210X.12066

8. M. Goberna and Miguel Verdu. Predicting microbial traits with phylogenies. The ISME Journal 10:956-967, 2016. doi: 10.1038/ismej.2015.171.

**Reviewer #3: **

The authors present an evolutionary model for gene annotation and use it with a Bayesian MCMC framework to infer unknown gene annotations and even indicate errors in known annotations. This paper is an effort towards automatic gene annotation which is an open and important problem in gene ontology. The paper has some very good features -- the authors explain the model in detail, talk about its drawbacks and possible extensions in the discussion, show the effect of change in prior on the final estimated parameters, show the effect of pooling data and sensitivity analysis and provide a github repository with the code and results. For these reasons I think the paper should be accepted for publication, although I suggest consideration for the following questions and suggestions for the final publication. I have marked the major and minor comments in the list.

1. [Major] The proposed method has not been adequately contrasted with existing methods.

For example it would be good to show how the probabilistic framework differs from the one used in SIFTER.

Moreover, why have the results not been compared to SIFTER?

2. [Major] The sensitivity analysis is a bit confusing.

I believe the analysis is done on the PANTHER phylogenies.

So if a parameter is shown to have negligible effect on the MAE with other parameter fixed -- is it due to how the model behaves or due to the kind of data that is used in the fitting process?

This sensitivity analysis should be repeated on simulated datasets to truly understand which parameters in the model are important for accurate predictions.

The paired t-test results mentioned at the end of Section 3.3 should also be shown in the paper.

3. [Minor] What are the system specifications for the run-time analysis between the proposed method and SIFTER at the end of page 12? `Regular laptop computer' is not informative enough. Please provide the processor used, its speed and RAM for the system.

4. [Minor] The discovery section looks at the solutions with high certainty in prediction of presence or absence of function.

On the other hand Featured Examples section presents one low accuracy result that reveals an area of inconsistency in current literature.

It would be helpful to describe other putative areas where the current knowledge might be inconsistent based on the predictions of your method.

**Have all data underlying the figures and results presented in the manuscript been provided?**

Reviewer #1: **No: **There is a GitHub repository (https://github.com/USCbiostats/aphylo-simulations) that looks like it should contain the spreadsheets, but I could not find them. It looks like there are scripts for producing the figures and tables, which is very helpful, but it does not seem to contain the actual spreadsheets (or I could not find them).

Reviewer #2: Yes

Reviewer #3: Yes

PLOS authors have the option to publish the peer review history of their article (what does this mean?). If published, this will include your full peer review and any attached files.

Reviewer #1: No

Reviewer #2: No

Reviewer #3: No
---

## [Decision Letter · Decision Letter 1]

7 Dec 2020

Dear Vega Yon,

Thank you very much for submitting your manuscript "Bayesian parameter estimation for automatic annotation of gene functions using observational data and phylogenetic trees" for consideration at PLOS Computational Biology. As with all papers reviewed by the journal, your manuscript was reviewed by members of the editorial board and by several independent reviewers. The reviewers appreciated the attention to an important topic. Based on the reviews, we are likely to accept this manuscript for publication, providing that you modify the manuscript according to the review recommendations.

Thank you for addressing the majority of issues raised by reviewers. Please address the remaining issues raised by Reviewer 1.

Sincerely,

Mohammed El-Kebir

Guest Editor

PLOS Computational Biology

Jian Ma

Deputy Editor

PLOS Computational Biology

[LINK]

Thank you for addressing the majority of issues raised by reviewers. Please address the remaining issues raised by Reviewer 1.

Reviewer's Responses to Questions

**Comments to the Authors:**

Reviewer #1: The authors made good-faith efforts to address the reviewer comments, and they have largely addressed my concerns. I have a few comments about their resubmission:

1. Each reviewer requested that the authors compare aphylo with existing methods, so the authors compared aphylo with SIFTER. This comparison required a substantial amount of effort, and the authors have helpfully shared their code publicly. However, the authors also noted that the comparison is somewhat awkward because aphylo and SIFTER have different objectives. I think that there is still an opportunity for more relevant benchmarking and for better clarifying aphylo's place in the computational landscape of phylogenetic approaches for gene function annotation. For example, should readers use aphylo instead of other methods in their current pipelines, should they use aphylo to choose parameters for other methods, etc.?

2. The authors say that users can perform large-scale analysis with a "regular" computer, but they later describe a computer with 20 CPU cores and 250 GB RAM. Are these the same or different computers? Can the authors unambiguously describe the system or system requirements for aphylo on the analysis in the paper?

3. It was not immediately clear what was changed and what was kept from the previous version of the manuscript because there were no highlights, etc. However, there are several opportunities to clean up the submission, particularly in the figures and figure lengths. For example, the y-axis for Figure 10 is "Value", and the y-axis for Figure 13 is "aphylo - SIFTER", so the authors could be more precise.

Reviewer #3: I am pleased with the reviewers addressing the major comments in the paper and would be happy to recommend acceptance of this paper.

**Have all data underlying the figures and results presented in the manuscript been provided?**

Reviewer #1: **No: **There is a GitHub repository (https://github.com/USCbiostats/aphylo-simulations) that looks like it should contain the spreadsheets, but I could not find them. It looks like there are scripts for producing the figures and tables, which is very helpful, but it does not seem to contain the actual spreadsheets (or I could not find them).

Reviewer #3: Yes

PLOS authors have the option to publish the peer review history of their article (what does this mean?). If published, this will include your full peer review and any attached files.

Reviewer #1: No

Reviewer #3: No
---

## [Editor Report · Decision Letter 2]

30 Dec 2020

Dear Vega Yon,

We are pleased to inform you that your manuscript 'Bayesian parameter estimation for automatic annotation of gene functions using observational data and phylogenetic trees' has been provisionally accepted for publication in PLOS Computational Biology.

Best regards,

Mohammed El-Kebir

Guest Editor

PLOS Computational Biology

Jian Ma

Deputy Editor

PLOS Computational Biology

---

## [Editor Report · Acceptance letter]

11 Feb 2021

PCOMPBIOL-D-20-00737R2 

Bayesian parameter estimation for automatic annotation of gene functions using observational data and phylogenetic trees

Dear Dr Vega Yon,

I am pleased to inform you that your manuscript has been formally accepted for publication in PLOS Computational Biology. Your manuscript is now with our production department and you will be notified of the publication date in due course.

With kind regards,

Alice Ellingham
